# Dendrohydrology and water resources management in South-Central Chile: Lessons from the Río Imperial streamflow reconstruction

Alfonso Fernández[1], Ariel Muñoz[2], Álvaro González-Reyes[3], Isabella Aguilera-Betti[2,4], Isadora Toledo[2], Paulina Puchi[2,5], David Sauchyn[6], Sebastián Crespo[2], Cristian Frene[7], Ignacio Mundo[8], Mauro González[9], and Raffaele Vignola[10]

[1]Departamento de Geografía, Universidad de Concepción, Concepción, Chile
[2]Instituto de Geografía, Pontificia Universidad Católica de Valparaíso, Valparaíso, Chile
[3]Instituto de Ciencias de la Tierra, Facultad de Ciencias, Universidad Austral de Chile, Valdivia, Chile
[4]Centro Transdisciplinario de Estudios Ambientales y Desarrollo Humano Sostenible (CEAM), Universidad Austral de Chile, Valdivia, Chile
[5]University of Padova, Italy
[6]Prairie Adaptation Research Collaborative, University of Regina, Saskatchewan, Canada
[7]Instituto de Ecología y Biodiversidad, Pontificia Universidad Católica de Chile, Santiago, Chile
[8]Instituto Argentino de Glaciología, Nivología and Ciencias Ambientales and Facultad de Ciencias Exactas y Naturales, Universidad Nacional de Cuyo, Mendoza, Argentina
[9]Instituto de Conservación, Biodiversidad y Territorio, Universidad Austral de Chile, Valdivia, Chile
[10]Cátedra Latinoamericana en Decisiones Ambientales para el Cambio Global, Turrialba, Costa Rica

*Correspondence to:* Alfonso Fernández (alfernandez@udec.cl)

**Abstract.** Streamflow in South-Central Chile (SCC, $\sim$37ºS-42ºS) is vital for agriculture, forestry production, hydroelectricity, and human consumption. Recent drought episodes have generated hydrological deficits with damaging effects on these activities. This region is projected to undergo major reductions in water availability, concomitant with projected increases in water demand. However, the lack of long-term records hampers the development of accurate estimations of natural variability and trends. In order to provide more information on long-term streamflow variability and trends in SCC, here we report findings of an analysis of instrumental records and a tree-ring reconstruction of the summer streamflow of the Río Imperial ($\sim$37º40'S-38º50'S). This is the first reconstruction in Chile targeted at this season. Results from the instrumental streamflow record ($\sim$1940 onwards) indicated that the hydrological regime is fundamentally pluvial with a small snowmelt contribution during spring, and evidenced a decreasing trend, both for the summer and the full annual record. The reconstruction showed that streamflow below the average characterized the post-1980 period, with more frequent, but not more intense, drought episodes. We additionally found that the recent positive phase of the Southern Annular Mode has significantly influenced streamflow. These findings agree with previous studies, suggesting a robust regional signal and a shift to a new hydrological scenario. In this paper, we also discuss implications of these results for water managers and stakeholders; we provide rationale and examples that support the need for the incorporation of tree-ring reconstructions into water resources management.

# 1 Introduction

Streamflow in South-Central Chile (SCC, ∼37ºS-42ºS) is vital for agriculture, forestry production, hydroelectricity, and human consumption (Lara et al., 2003; Rubio-Álvarez and McPhee, 2010). With more than 55% of Chilean agriculture and forestry production delivered from this region (Instituto Nacional de Estadísticas, 2007), the drought episodes that occurred in the last few decades and the associated hydrological deficit have had damaging effects (Garreaud, 2015). These drought episodes have been linked to a significant decreasing trend in regional precipitation (Pezoa, 2003; Aravena and Luckman, 2009; González-Reyes and Muñoz, 2013), with amounts 40% below the 1901-2005 mean (Trenberth et al., 2007). SCC is expected to undergo important climate changes. Analysis of the multi-model ensemble for the scenario RCP4.5 presented in the Atlas of Global and Regional Climate Projections (IPCC, 2013) indicates 10 to 30% reduction in spring and summer (October to March) precipitation by 2016-2035 and 2046-2065 relative to 1986-2005. The same projection forecasts 0.5 to 2ºC warming for summer (December- February). Drier and warmer summers may make SCC more vulnerable to water scarcity, given that this is the season of highest water demand in this region (Garreaud, 2015).

Current and projected increases in water demand in this region is likely a source of high uncertainty for future scenarios of water use (Lara et al., 2003). However, the lack of long-term observational records of hydrometeorological variables makes difficult the development of accurate estimations of natural variability, useful for determining the severity of the recent observed deficit in streamflow (Lara et al., 2008). Therefore, there is need to extend the observational record to better understand long-term (e.g. centuries to millennia) variability and trends in streamflow, thus providing useful information for hydrological assessment of governmental and private planning initiatives within watersheds of SCC (Lara et al., 2015; Muñoz et al., 2016).

The Río Imperial, a major river in SCC, drains an area of 12,763 km$^2$, extending between ∼37º40'S to ∼38º50'S (CADE-IDEPE, 2004) (Fig. 1). The river begins at the confluence between the Chol Chol and Quepe rivers, in the ninth Chilean administrative region known as "Araucanía". Summer streamflow is controlled by rainfall, as most snowmelt dominated discharge (usually occurring in late austral spring) has vanished (CADE-IDEPE, 2004). This river is currently utilized for irrigation, fishing, tourism, and transportation (in certain sections). For instance, in 2001 4% of the basin area was potentially usable for irrigation agriculture, with some projections suggesting that this proportion will grow to ∼10% (Ayala-Cabrera y Asociados, 2001). Additionally, the Río Imperial has high hydroelectric potential, ranked eleventh in the country, with 455.8 MW (Santana et al., 2014). An emerging sociohydrological problem in this basin is related to the allocation of water rights by the government: there is more streamflow allocated than the current river discharge. What seems to prevent a water availability crisis is that not all the allocated water is being exploited (Ayala-Cabrera y Asociados, 2001). Facing scenarios of water scarcity may obligate some users to fully claim their rights, likely triggering a regime shift with uncertain implications. However, the ability to ponder the possible consequences of these predicted changes is limited because accurate calculations of extreme scenarios are essential but largely nonexistent. For this reason, long-term evaluations of Río Imperial discharge are urgently needed.

Tree-ring analysis has proven to be a useful tool for supplementing available observations of streamflow. These proxy records can provide time series at yearly resolution, capable of delivering historical information on natural variability, estimations of return periods of extreme events such as droughts, and on the correlation with large-scale climatic forcings (Meko and

Woodhouse, 2011; Sauchyn et al., 2011). In Chile, dendrohydrology has already been utilized for streamflow reconstruction in four major rivers: Maule ($35°30'$S, (Urrutia et al., 2011)), Biobío ($37°10'$S, (Muñoz et al., 2016)), Puelo ($41°39'$S, (Lara et al., 2008)), and Baker ($47°40'$S, (Lara et al., 2015)). To date, however, these studies have not yet been fully utilized for watershed planning and management. In order to provide more information on long-term streamflow variability and trends in SCC and thus contributing to improving water resource management in this key region for the country's economy, here we report findings of a tree-ring based reconstruction of the summer streamflow of Río Imperial. This is the first reconstruction of streamflow in summer, the season with the highest water demand. Our research focused on expanding current understanding of long-term streamflow, thus providing more evidence on both regional coincidence among studies and seasonal/local particularities in streamflow. Therefore, our objectives were to (a) determine whether or not current summer streamflow changes in the Río imperial are unprecedented in the multi-century scale; (b) establish if the return period of extreme years of high and low streamflow has changed in the last few centuries; and (c) estimate the correlation between the multi-century variability of the Río Imperial streamflow and climate modes of natural variability such as El Niño Southern Oscillation (ENSO), the Southern Annular Mode (SAM), and the Interdecadal Pacific Oscillation (IPO). Given that this is the fifth streamflow reconstruction for a Chilean river and that available results from previous studies have yet to be utilized for water resources planning, in our discussion section we also (a) summarize arguments related to physical processes that may support our summer reconstruction as a more robust time series compared to some of the previous studies, and (b) in light of our results, we evaluate strengths and weaknesses of a recently enacted method to calculate minimum ecological discharge, hoping to engage authorities, managers and stakeholders in considering proxy-data for improving water resources management and research.

## 2   Methods

### 2.1   Analysis of Instrumental Streamflow Records

In this study, January and February represent summer streamflow, the period of the year with the streamflow is closest to the river's baseflow and when annual precipitation is at its lowest (Fig. S1). Streamflow during these months is sensitive to changes in soil moisture and thus summer precipitation (or the lack of) during that period or from previous months. We selected three instrumental records (Fig. 1) representing the natural streamflow regime of the mid and lower sections within the river basin (Table 1) for the period 1947-2015. These stations gauge rivers where the water has not been diverted for irrigation and hydroelectricity and are also located in two protected areas: Malalcahuello National Reserve and Conguillío National Park. Flow at downstream stations is highly conditioned by engineered structures (e.g. water intakes). We tested the "natural regime" represented by these records by conducting a correlation study between these stations and other hydroclimatic variables such as rainfall. We utilized double-mass curves to determine the length of the time series for calibration of the reconstruction model (Muñoz et al., 2016), discarding records that did not follow rainfall trends; this resulted in the elimination of a few years in the earliest section of the time series.

## 2.2 Reconstruction and Analysis using Tree-ring Chronologies

We built a multiple regression model relating three tree-ring chronologies from *Araucaria araucana* to records of Río Imperial's summer streamflow (Fig. 1 and Table 2). These chronologies were those showing the highest correlation with streamflow records, extracted from an initial group of 19 chronologies taken from *Araucaria araucana* (16) and *Austrocedrus chilensis* (3). We utilized the softwares ARSTAN (Cook, 1985) and COFECHA (Holmes, 1983) for standardization and dating of the tree-ring chronologies. This standardization procedure applies linear and exponential statistical models to suppress temporal auto-correlation in the series. These statistical procedures minimize the differences in tree-ring width between individual trees (e.g. due to local dissimilarities in soil moisture or differences in age) and therefore maximize our ability to retrieve the common tree-ring pattern related to the hydroclimate.

The three available instrumental records for the basin were transformed into standardized anomalies and then averaged to constitute a single composite time series (Lara et al., 2008; Muñoz et al., 2016). To obtain a valid simulation of the streamflow from the tree-ring chronologies, we employed the "leave-one-out" regression technique using the period 1947-2005 as the validation window, testing several predictors from the three tree-ring chronologies. The predictors consisted of chronologies for the growth year $(t)$, as well as backward and forward lags of one and two years $(t-2, t-1, t+1,$ and $t+2$, respectively). We found that the most robust statistical model corresponded to the following:

$$IS = 0.07 - 2.05PIN_{t+1} + 2.83LYV_t + 2.69PAG_{t-1} - 1.46PAG_{t+2} - 1.97LYV_{t-1} \tag{1}$$

Where $IS$ is the reconstructed summer streamflow for the Río Imperial, while $PIN$, $LYV$, and $PAG$ correspond to the tree-ring chronologies as described in Table 3. The forward lagged time series included suggest that the statistical model has predictive skill, as shown in previous studies (e.g. Sauchyn et al. (2015a)). In this model, LYV and PAG correlated positively with same-year streamflow (0.38 and 0.43, respectively), with that of the previous year (0.53 and 0.41), and with streamflow recorded two years before (0.3 and 0.23). Conversely, PIN correlated negatively with streamflow recorded in previous years (-0.56 for the first and -0.38 for the second previous year). We conjeture that the negative correlation between PIN and the instrumental data is related to features of the sampling site, located in a narrow sector with recurrent fog, where atmospheric moisture and precipitation in summer may produce a relative reduction in incoming radiation and temperature; these conditions may reduce the rate of the tree-ring growth. Mundo et al. (2012) reported similar findings for PIN when analyzing as larger group of chronologies.

During the verification period we utilized (a) the $R^2$ ($R^2$adj) to assess the explained variance; (b) the Reduction of Error (RE) statistic to account for the relationship between the actual value and its estimate; (c) the F statistic for assessing the accuracy of the regression; (d) the Root Mean Square Error (RMSE) as well as the Standard Error (SE) as measures of uncertainty; (e) the Durbin-Watson test for determining the degree of auto-correlation of the residuals; and (f) the Variance Inflation Factor (VIF) to check the possibility of multicollinearity in the regression (see Ostrom (1990) for details). The Variance Inflation Factor (VIF) evaluates the multicollinearity of the predictors: a VIF close to 1 means a low or no multicollinearity (Haan, 2002) while a value above 10 is associated with multicollinearity problems between predictors (O'brien, 2007).

For studying the return period of extreme low flows in the streamflow reconstruction, we applied the peak over threshold (POT) approach, using the low flows corresponds to a threshold $\leq 20^{th}$ percentile. We also estimated the recurrence rate of drought events utilizing a kernel estimation technique with a Gaussian function and 50-yr bandwidth. The kernel-based estimation of drought recurrence allows for detection of nonlinear and non-monotonic trends without imposing parametric restrictions. Furthermore, a smooth kernel function produces more realistic estimation of drought recurrence. We calculated

5    a confidence interval at the 95% level based on 1000 bootstrap resampling steps (Cowling et al., 1996) to estimate bias and variance properties of drought recurrence in the reconstruction. The kernel estimation, bandwidth selection, and bootstrap algorithm were computed in the free R Project platform software (R Core Team, 2016). We assessed frequency of cycles in the reconstruction by applying the following methods: Blackman-Tukey (Ghil et al., 2002), Multi-taper, Singular Spectral Analysis, and a Continuous Wavelet Transform Analysis.

10    We further performed a number of statistical analyses comparing observed, reconstructed streamflow time series, and other relevant hydroclimatic time series. For the period with available instrumental data (second half of the 20$^{th}$ century), we calculated Pearson correlations between these records and the Temuco rainfall gauge (TEM in Fig. 1; 38º46'S - 72º38'W), one of the few continuous records available in this watershed. We further analyzed the coherence of dominant periods of streamflow between the instrumental record and the tree-ring reconstruction, derived from a Singular Spectral Analysis ((Lara et al., 2015)

15    and references herein). In addition, we compared the summer streamflow reconstruction of the Río Imperial with a November-December rainfall reconstruction of North Patagonia (Villalba et al., 1998). These chronologies are statistically independent, since the latter derives from samples extracted from *Austrocedrus chilensis*. This comparison is justified by the fact that studies in other watersheds in the SCC region have shown relatively fast response of the discharge to changes in precipitation (e.g. Zúñiga et al. (2012)). Finally, we studied the correlations between observed/reconstructed streamflow and time series represent-

20    ing modes of regional climate variability, namely El Niño Southern Oscillation (ENSO), the Interdecadal Pacific Oscillation (IPO), and the Southern Annular Mode (SAM). For the ENSO, we utilized the Southern Oscillation Index (SOI) provided by the Climate Analysis Section, Climate and Global Dynamics Laboratory at NCAR[1]. This SOI is computed from monthly mean sea level pressure anomalies at Tahiti and Darwin and has became standard for climatic studies (Trenberth and Caron, 2000). Negative SOI anomalies correspond to relatively warmer conditions in the Eastern Pacific Ocean (or more El Niño-like

25    conditions). The IPO time series is from the reconstruction presented in Vance et al. (2015). For the time series representing the activity of the SAM, we selected the Antarctic Oscillation (AAO) index, defined as the leading principal component of sea level pressure in the region south of 20ºS (Thompson and Wallace, 2000) and also calculated from the NCAR-NCEP reanalysis[2].

---

[1]http://www.cgd.ucar.edu/cas/catalog/climind/soi.html (last accessed 03/12/2018)

[2]For details, go to http://jisao.washington.edu/data/aao/slp/ (last accessed 04/17/2017)

## 3 Results

### 3.1 Streamflow Trends and Variability from the Instrumental Record

Our analysis of instrumental data corroborated previous studies where the regime of the Río Imperial was characterized as fundamentally pluvial with a small snowmelt contribution during spring (CADE-IDEPE, 2004). The hydrographs in Fig. 2 show that around 40% to 50% of the streamflow occurs between June and August, although at the Cautín station spring months contribute slightly more to the overall flow relative to the other stations. Conversely, summer streamflow represents 5% to 10% of the total yearly amount. We also found a high correlation, as high as 0.61 for TEM, between summer discharge and rainfall of the same season as well as with rainfall observations of previous months.

All observations showed a decreasing trend since the beginning of the records, both for the annual as well as for the summer mean (Fig. 2). Four of five lowest discharge records occurred after 1995, notably 1999, whereas the five highest are detected before 1981, with four of them before 1974 (Table 2). We calculated the proportional departures of each of these years (in percentage) relative to the long term average, finding that the five lowest-flow years showed little spread, with a mean departure of 54.06% ±6.14 (1 standard deviation), whereas for the five highest-flow ones there was more spread (74.12% ±23.49). Furthermore, the record suggested that high flow years tended to be more extreme (relative to the mean) than low flows.

### 3.2 Streamflow Reconstruction: Features and Interpretation

Two of three *Araucaria araucana* tree-ring chronologies extended 800 years or more (PAG and LYV; Fig. 3 and Table 3), which corroborates findings from a previous study (Mundo et al., 2012) on the potential of this species for providing long paleoclimatic reconstructions. Combining the resultant common period among the tree-ring chronologies (1606-2005) with the Expressed Population Signal (EPS) statistic yielded a 296-year (1709-2005) time series of reconstructed summer streamflow. Although the EPS for the PIN chronology was 0.85 only after 1750 (Table 4), the large number of tree-growth series in the PAG and LYV chronologies (20 and 34 by 1709, respectively, see Fig. 3) allowed for the reconstruction to begin in 1709. An EPS greater than 0.85 for a given tree-ring chronology is often assumed as proof of its reliability, because it indicates that at least 85% of the chronology variance corresponds to a common signal (Wigley et al., 1984). Additionally, the VIF for the reconstruction period had values close to 1, especially for the PAG chronology, indicating that the regression model had no problems of multicollinearity (Table 4). From Table 4 we also noted that the chronologies presented high autocorrelation, consistent with the fact that tree-growth has a temporal memory associated with trees' capacity to store carbon and water, and the uptake rate of soil moisture remaining from previous seasons and years. Some of the statistical procedures applied to the tree-ring chronologies are meant to minimize these effects, but it is virtually impossible to eliminate all. However, when calculated for the reconstruction the autocorrelation reduced to 0.25 which, although high considering that the instrumental record is essentially free from autocorrelation (-0.093), it is nevertheless better than the individual chronologies. We extended this record using the composite instrumental record and the variance of the reconstruction. This way, the reconstruction allowed us to cover and assess the hydroclimatic significance of the latest anomalous dry period detected for Central Chile (2011-2015), referred to as "megadrought" (Garreaud, 2015).

Comparison between standardized anomalies of the reconstruction and observations during the calibration period (1947-2010) suggested a high reconstruction skill for summer streamflow record (Fig. 4 top-left panel). We also computed correlations between observed/reconstructed streamflow and TEM for each month of the previous year, finding significant figures for December and February of the previous year (p-value < 0.1 and < 0.05, respectively) in the case of the instrumental record; for the reconstruction, the correlation is significant for June and December (p-value < 0.05, see details in Fig S3). In addition, an analysis of coherence between dominant periods showed that both time series presented similar cycles of three, five, and 30 years (Fig. 4 top-right panel). This good performance was further corroborated with these statistics: (a) $R^2$adj=0.47, indicating that the regression model explained 47% of the variance in the predictand; (b) small errors, with RMSE=0.71, RE=0.36, and SE=0.66; (c) a statistically significant model as reflected by the F statistic (10.08, p<0.0001); while (d) the residuals were not auto-correlated (Durbin-Watson = 2.25, p-value < 0.01). Previous studies have found similar performances for reconstructions in other regions of the world (e.g. Woodhouse (2001)), which lead us to assert that our reconstruction is a reliable representation of the Río Imperial's summer streamflow. Another proof is the detection of significant agreement between the reconstruction and observations for the period 1980-2010 for the region $\sim$35ºS-42ºS (Fig. S2).

We evaluated trends and frequency of extreme events for the whole reconstructed period. This analysis revealed a negative, although no statistically significant trend for the reconstruction. What we did observe was a low frequency of extreme flows, although during most of the 20th century ($\sim$1910-1970) high flows (above the percentile 60) clustered. On the other hand, low flow periods (below the percentile 20) occurred more frequently in the late 1800s, early 1900s, and after 1980. Importantly, streamflow below the historical average characterized the post-1980 period (Fig. 4 lower panel). The application of the Blackman-Tukey method allowed us to detect high frequency cycles (2-7 years) and mid-to-low frequency (> 8 years). The Multi-taper method and the Singular Spectral Analysis revealed that a $\sim$4-year cycle captures 10% of the variance, a $\sim$7-year cycle captures 7%, while we also found a 16-20-years cycle, corresponding to 20% of the variance. This last long cycle was also found when applying a Continuous Wavelet Transform Analysis, in this case a significant 16-32-years frequency between 1800-1950; higher frequency cycles occur along the whole period but appear more significant after 1900 (Fig. S4).

In order to assess the uniqueness of this recent period of extreme summer streamflow, we (a) divided the tree-ring reconstruction into continuous periods of one, five, 10, and 20 years; and (b) we ranked those periods according with their departure from the mean. Although somewhat arbitrary, the length of these windows are defined as a mean for providing context for the occurrence of extreme flows along the whole study period in way that is easier to understand for water managers. According to this classification, the dry period 1996-2000, one of the driest in the instrumental record, ranks fourth in the reconstruction, closely followed by 1987- 1991. Results also indicated that the lowest flow events since the mid 20th century are somewhat relevant at the five-year scale. However, the period 2011-2015 did not show low flows as extreme as other periods in the reconstruction; it only ranked high in the instrumental record. At the 10-year scale, post-1980 low discharges ranked only third and fourth. The most striking relevance of modern low flow appeared at the 20-year scale, where the period 1986-2005 ranked first, suggesting this period as the driest since the beginning of the reconstruction. The same analysis for extremely high streamflow showed that the mid 20th century was persistently ranked among the five in all the timescales considered (Table 5).

Since 1980, years in the lowest 20[th] percentile of the reconstruction have become more frequent. We calculated the return period of these low flow years in different periods of the reconstruction. We found (a) that during 1709-1750 and 1940-1960, events with streamflow below the 20[th] had a 20-year return period; (b) a 5-year return period in 1750-1880; (c) a predominantly 2 to 3-year period for 1880-1930; and (d) a trend toward a 2-year return period since 1960. In addition, we found that since ∼1980 more droughts below the 10[th] percentile are also becoming more frequent (Fig. 5).

We found the Río Imperial reconstruction and the precipitation reconstruction for North Patagonia (Villalba et al., 1998) to be significantly correlated. Closer inspection of the year to year correspondence between these two reconstructions (Fig. 6a) revealed that the interannual variability was somewhat different, especially for the most negative departures in which there was ocasional coincidence. A 30-year spline filter shows the similarity at lower frequency between these two time series, although with noticeable lags (nevertheless irregular in length) between peaks and troughs, as for example in the period ∼1730-1740 (Fig. 6b). Overall, the reconstructions coincided in the number and length of high and low periods.

### 3.3 Streamflow Correlation with Regional Climate Modes of Variability

The instrumental discharge record correlated with the ENSO and the SAM (Table 6). While summer discharge had a Pearson correlation of 0.32 with the SOI from February to March of the previous year, the coefficient was as high as -0.66 with the AAO of the previous 12 months (September to August). The AAO-streamflow correlation of streamflow with the instrumental record and tree-ring reconstruction were identical when calculated for the period September-December of the immediate previous year. This correlation was weaker (-0.3) when we removed the linear trend, indicating that a significant portion of the streamflow trend is shared with the trend in the AAO. The agreement for September-December, as well as the relatively similar correlations observed for February-March (instrumental record) and March (tree-ring), reaffirmed the previous finding about the high skill of the reconstruction to reproduce essential properties of the instrumental record. An additional confirmation of the influence of the SAM on streamflow can be observed in Fig. 6, where we found a similar correlation between a long-term reconstruction of the AAO (Villalba et al., 2012) and our streamflow reconstruction, with a statistically significant Pearson r of -0.33 for both high and low frequencies. In order to discard the possibility of a non-significant correlation, we performed further correlations between the AAO and a pre-whitened version of (a) the reconstruction and (b) the observations (removing the lag-1 autocorrelation). Results reaffirmed the negative and statistically significant correlation, with -0.287 (p-value=0.033) for the reconstruction and and -0.29 (p-value=0.029) for the observations, respectively. In the case of the IPO, we found a statistically significant negative correlation between that time series and the 16-20-year cycle of our reconstruction (-0.38, n=295), although it looks more coherent during the 20th century.

### 4 Discussion

The tree-ring reconstruction for the Río Imperial has revealed new information and insights on long-term streamflow variability, thus allowing for the assessment of changes in summer water availability in the SCC region, including the long-term significance of recent extreme hydrological events throughout the region. This streamflow reconstruction corresponds to the

fifth record for Chilean rivers, but the very first focused on summer streamflow. Some of our findings agree with those of previous studies, which strongly suggests that our reconstruction represents well the summer hydroclimate of a large region in SCC. This is manifested by the good skill our reconstruction in simulating instrumental records along the wider region 35°S-42°S (Fig. S2), which further supports findings by González-Reyes et al. (2017), who stated that this region configures a hydroclimatic cluster.

The noticeable decreasing trend post-1980 observed in the instrumental record as well as in our reconstruction, which was to a certain extent identified in all previous reconstructions (Lara et al., 2008, 2015; Urrutia et al., 2011; Muñoz et al., 2016), corroborates the occurrence of an ongoing and unprecedented long-lasting summer low flow in SCC, which also includes the recent drought detected for Central Chile (Garreaud, 2015). We note that the post-1980 period represents ~50% of the composite instrumental time series, while it only represents ~10% of the reconstructed period; we posit this highlights the uniqueness of the post-1980 period. Thus, this period is noteworthy as the one with the lowest summer streamflow of the whole reconstruction. This low flow period gradually emerged as the most acute as the time window was broadened (Table 5), despite the fact that no year during this timespan ranked high in the lowest percentile of the reconstruction. As shown above, the decreasing trend observed in our results has already been detected in other rivers of the region, specifically south of ∼37°30'S (Masiokas et al., 2008; Rubio-Álvarez and McPhee, 2010). The implications of this finding for water management and planning purposes are related to the ability to detect the effect of regional climate changes in the area. The increasing number of low flow years may indicate a regime shift in the summer dynamics, which can be exacerbated with predicted increased warming and precipitation reduction in SCC (Garreaud, 2015). Our results indicated a decreasing trend in summer streamflow with a smaller natural variability when compared with the rest of the tree-ring record.

Our results additionally posited the SAM as the most influential climate mode driving the variability of the Río Imperial's summer streamflow, especially during the second half of the 20[th] century. The correlation we found between SAM and the reconstruction establishes the tree-ring time series as independent new proof of the importance of the SAM in modulating precipitation and streamflow in a large region of Central and South of Chile (Villalba et al., 2012; Muñoz et al., 2016). The SAM has been extensively described as a strong modulator of precipitation variability in SCC (Aravena and Luckman, 2009; Villalba et al., 2012; González-Reyes and Muñoz, 2013). The tree-ring reconstruction showed a statistically significant correlation with the SAM reconstruction presented in Villalba et al. (2012), especially for the long-term trend; we consider this result confirms that the reconstruction of summer streamflow for the Río Imperial captures characteristics of the regional hydroclimate. The reconstruction featured in Villalba et al. (2012) contains two of the tree-ring chronologies presented in our study (PIN and LAN), in a pool of more than 1000 series developed from different species of the Southern Hemisphere. Our summer stream-flow reconstruction is key in further supporting the SAM-streamflow relationship, because in this season and overall, snowmelt is a minor contributor to discharge (see Fig. 2). This makes rainfall the principal source of water for the river. During the last 75 years or so, the SAM has been shifting to a positive phase, unprecedented in the last 1000 years (Abram et al., 2014). A positive phase means that the pressure gradient between the midlatitudes and the polar latitudes is negative, that is, more positive or less negative in the midlatitudes relative to polar latitudes. This gradient shifts the core of the southern westerly winds towards the south, limiting the effect of storm tracks over midlatitudes. In the past, the SAM has been shown to be negatively correlated

with the frontal activity and hence precipitation amounts, especially for the spring season (Silvestri and Vera, 2003). Given the relatively short delay between precipitation and discharge in the region (e.g. Zúñiga et al. (2012)), this relationship may well explain the decreasing summer streamflow trend observed for the last decades, as detected in the September-December negative correlation between the AAO in relation to both instrumental records and the tree-ring reconstruction (Table 5). This way, the post-1980 trend in summer streamflow of the Río Imperial may be linked to the aforementioned positive phase of the SAM (Abram et al., 2014). More so, this close relationship could mean an unprecedented decreasing trend in summer streamflow in the Río Imperial for the last millennia. However, it is important to consider results according to Blázquez and Solman (2016) on the relationship between precipitation, frontal activity, the ENSO, and the SAM in the Southern Hemisphere. In that study, ENSO (defined as the second EOF from monthly anomalies of the 500 hPa geopotential level) correlated higher with frontal activity in spring at the interannual timescale, although the correlation was also statistically significant with the SAM (defined as the first EOF from monthly anomalies of the 500 hPa geopotential level). Blázquez and Solman (2016) also demonstrated that precipitation in SCC is significantly correlated with frontal activity. These studies suggest that the impact of the SAM on precipitation in the study area can also be associated with ENSO dynamics. For example, the low flow for the summer 1999 was concomitant with a high AAO and and a strong La Niña event in 1998-1999. Recent studies have found that in-phase (e.g. positive ENSO and positive SAM) and out-of-phase occurrences of these climate modes can lead to varying strength of the associated atmospheric circulation (Fogt et al., 2011; Wilson et al., 2016), possibly controlling the intensity of their impacts on precipitation and hence streamflow. In fact, Quintana and Aceituno (2011) in a study analyzing the changes in rainfall in Central and South Central Chile found that ENSO and AAO are not independent modes of variability and rather they both affect precipitation regimes. It is also important to consider possible different seasonal effects of these modes of variability, as for example findings in Barria et al. (2017), who utilizing a 11-year moving average of a 300-year tree-ring streamflow reconstruction for the upper Biobío river determined a negative correlation between with the Pacific Decadal Oscillation (PDO) and streamflow during the snowmelt season (October-March) while a positive correlation between the SAM and that seasonal streamflow. Thus, we believe that the SAM-streamflow statistical relationship we found should be tested further in order to develop more accurate predictive models of extreme interannual dynamics. Using output from the CMIP5, Lim et al. (2016) predicted that the positive phase of the SAM will continue in some climate change projections, likely (but not certainly, given the difficulties in simulating the ENSO-SAM interactions) impacting precipitation in the midlatitudes. In this scenario, the SAM could provide more accurate statistical models of return periods.

Another relevant aspect of our reconstruction is that, like previous studies, it seemed to better represent low flow conditions rather than high flows. Urrutia et al. (2011) discussed this finding in their reconstruction of the Río Maule arguing, following Villalba et al. (1998), that this was likely a consequence of a smaller sensitivity of tree growth to precipitation above a certain threshold. However, the $R^2$adj of previous studies do not completely support that argument as the highest $R^2$adj (54%) was found in the Río Baker streamflow reconstruction (Lara et al., 2015), the southernmost and the wettest landscape of all sampled locations in Chile. In terms of $R^2$adj, our reconstruction ranks second (47%), followed by Biobío (45%, Muñoz et al. (2016)), while Puelo Lara et al. (2008) and Maule Urrutia et al. (2011) rank together with the lowest explained variance (42%). Notice that the two highest ranked reconstructions represent summer to fall streamflow, when the soil moisture is at its lowest in

the annual cycle. Elshorbagy et al. (2016) summarized empirical and theoretical arguments to better understand, on the one hand, the physical process linking tree growth and streamflow and, on the other hand, what determines whether tree-ring reconstructions reproduce dry conditions. In that study, the authors presented data from the Oldman River Basin ($\sim$49°40'N), a cold semi-arid watershed in western Canada. They suggested the likelihood of spurious self-correlation in the statistical models relating streamflow and tree-rings because precipitation may be considered as a confounding factor. The mechanism these authors suggested is that soil moisture modulates the relationship between tree growth and other hydrometeorological variables (e.g. evaporation or runoff), more so in areas (and perhaps seasons) where (when) soil moisture is relatively constant. Following these arguments, one should expect streamflow reconstructions to better represent years or periods of water scarcity rather than high flows. Elshorbagy et al. (2016) indeed found higher correlations between low flow and tree-ring width in the Oldman River basin. Our analysis, and the fact that the two of the previously reported reconstructions for Chile with the highest $R^2$adj were aimed to include summer months, suggest that reconstructions focusing on summer streamflow (at least partially as the case in Lara et al. (2015)) have physical meaning and not just statistical significance. This renders the Río Imperial reconstruction as a representative time series of summer streamflow for the last 301 years, with a high capacity for capturing occurrence of dry years. However, further research is needed because the Río Puelo reconstruction (Lara et al., 2008), also aiming summer to fall streamflow, did not rank as high as our reconstruction and the one for Río Baker.

Given the crucial role that summer streamflow plays for a number of economic activities, such as agriculture and human consumption in the basin, the tree-ring reconstruction we have presented here can provide water managers and stakeholders with improved support for decision-making and regional planning. For example, the post-1980 generalized regional decrease in streamflow, both instrumental (Rubio-Álvarez and McPhee, 2010) and reconstructed (e.g. Lara et al. (2008)), paralleled paradigmatic legal and institutional changes in national water management, whereby the policy to assign water rights (and hence rights to use a proportion of the streamflow) transited into a model "with strong private property rights, broad private economic freedoms, and weak government regulation" (*verbatim* Bauer (2004)). There is growing evidence of increasing water conflicts in SCC despite recent reform in this legal framework of water rights allocation (Bauer, 2015). Some have argued that at least part of the problem resides in the insufficient hydrological information for users, managers, and decision makers (Donoso, 1999). The relevance for water management in Chile is related to the mechanism for granting water rights, since it relies on the average river flow. By using this system, water rights have often exceeded the actual flow of rivers, specifically for the summer, forcing authorities to decree zones of water scarcity[3]. With the information obtained from tree-ring reconstructions, it is feasible for authorities to anticipate scenarios of future crises, particularly in drought cycles such as those described here. In this regard, a recently enacted (and modified) official decree defined that the minimum ecological discharge as the lowest 50% of the 95% confidence interval below the average discharge, determined from a probability distribution using a period of at least 25 years of instrumental observations[4]. The minimum ecological discharge is the current measure to determine when a given stream is so depleted that no more water rights must be allocated. Certainly, the 25-year window reflects the limited number of long-term records to reliably characterize throughout the country, obligating authorities to assume that the hydroclimatic

---

[3]An example of a decree for watersheds in the north of Chile can be found here https://goo.gl/S3n8d3 (last accessed 04/17/2017).

[4]See modification to decree Nº12, 2013, Ministerio del Medio Ambiente, Chile, available at https://goo.gl/uWyH9R (last accessed 04/17/2017).

regime of a given river is captured by a relatively short instrumental record. Interestingly, the text of the decree may likewise allow for the incorporation of tree-rings to inform the process of water rights allocation, since it actually (a) establishes the 25-year term as a minimum criteria for using available records to define the average river flow and (b) empowers the Chilean National Water Authority to incorporate other methods for river flow determination. On the other hand, this new decree defines explicitly that streamflow information has to be retrieved from an intake point, potentially limiting the use of tree-ring data because they usually represent hydroclimatic variability at the watershed scale. We argue that information derived from the multi-centennial view of streamflow that tree-rings provide can help to better contextualize calculations from instrumental records, providing an additional "watershed" or regional perspective on water management. Our results indicated that the years of lowest discharge from the instrumental record did not classify as the most extreme droughts when contextualized in the longer tree-ring reconstruction. This suggests that, for instance, the drought in 1998-1999 was not as extreme as other events in the past 301 years. Episodes such as the 1968-1969 (which in the case of the decree would need a period of at least 48 years to be detected) and 1998-1999 droughts impacted the regional economy with national repercussions, such as a 40% reduction in electricity supply (Fischer and Galetovic, 2001). Another example is the dry period after 2010 as a (Garreaud, 2015; Boisier et al., 2016), where our reconstructed streamflow indicates the that period was not as extreme as other that occurred in the last 301 years in SCC. In fact, our analysis indicates that 1996-2000 was a drier period in the reconstruction, although it only ranked fourth in Table 5. This finding implies that persistent rain deficits or meteorological droughts do not necessarily translate into extreme low streamflow because the storage capacity of certain hydrological basins can buffer these effects. We argue that, in order to understand the impacts and extent of extreme events such as droughts, tree-ring reconstructions from watersheds representing diverse hydroclimatic gradients, landcover characteristics, and geomorphometry are sorely needed.

## 5 Conclusions

The significance of our findings for planning is related to the need to consider a long-term view of the natural variability that defines the discharge regimes across the region. Planning governmental responses for periods of low flow based upon the one, two, or five-year events detected in the available instrumental record may lead to ineffective mitigation actions, because those events may not be the most extreme on record. Considering the implications of our results, we assert that the extended record we have provided can be utilized as a complementary source of information for water resources managers to more accurately determine the range of "worst-scenario" or "more-recurrent" droughts, thus helping the planning and implementation of more efficient mitigation policies. There are several examples where dendrohydrology has provided key information for decision-making process. Sauchyn et al. (2015b) presented an example in which a series of hydroclimatic scenarios derived from tree-ring chronologies were formally included in a government-supported hypothetical decision-making exercise where relevant stakeholders were tasked with developing adaptation measures for drought episodes. In that paper, and in the context of Canadian water policy, these authors stated that dendrohydrological research is a legitimate source of information for water resource planning and management. Another study, aimed to characterize long-term Athabasca River streamflow in Alberta, Canada, found that whereas a trend analysis of instrumental data depicted declining regional flows, a tree-ring reconstruction

showed periods of severe and prolonged low flows not captured by the instrumental record (Sauchyn et al., 2015a); suggesting that worst case scenarios estimated from historical gauge data likely underestimate the potential magnitudes of natural droughts in that region (Coulthard et al., 2016). Meko and Woodhouse (2011) described another application, where the Denver Water Board, the oldest and largest water supplier in the State of Colorado, USA, wanted to predict drought scenarios using reconstructions capable of capturing an episode that occurred during the 1950s as well as other more recent droughts. Gangopadhyay et al. (2015) asserted that dendrohydrology can help in determining drivers of periods of extreme hydrological episodes in the Upper Colorado River basin.

The evidence presented in our study regarding the relatively dry post-1980 period, the higher frequency of droughts detected in the present for the Río Imperial relative to previous centuries, and the SAM projected to follow a positive phase leading to less precipitation altogether suggest that the regional hydrology is moving toward a new regime of more frequent (although not necessarily more intense) droughts. In a similar manner as documented in several recent studies targeting other regions of the world (e.g Woodhouse and Lukas (2006) and Sauchyn et al. (2015b), among others), our reconstruction of natural hydroclimatic variability of the Río Imperial can provide a valuable framework to ponder the impact of anthropogenic climate change on future hydroclimatic changes in SCC, thus leading to a prudent water resource management strategy. Furthermore, water managers could use the reconstructed flows of the Río Imperial to plan mitigation strategies for drought events with return periods of five, 10 and 20 years. As this hydrological application of dendrochronology becomes widely accepted as valid and water managers realize that this kind of historical information can guide operational strategies in the forthcoming decades (Biondi and Strachan, 2012), the possible regime shift needs to be incorporated into adaptation and mitigation plans associated with the regional impacts of future global climate changes. This will avoid, or at least mitigate, negative consequences of these changes on the management of water resources for municipal and agricultural water supplies, electrical power, ecological habitats (e.g. Carrier et al. (2013)) and impacts on the economic activities in the region as water sources become stressed.

*Code and data availability.* Codes and data associated with this work can be accessed by emailing the first three authors.

*Competing interests.* The authors declare no competing interests

*Acknowledgements.* We first want to acknowledge the reviewers for their thorough assessment of our paper; we believe their comments have contributed to a better work. Support for the researchers involved in this work was provided by the following sources: (a) Chilean Research Council (FONDECYT 11161061, 11160454, 1171065), the Center for Climate and Resilience Research (CR)2 (FONDAP 15110009); (b) Proyectos Internos Pontificia Universidad Católica de Valparaíso (039.353/2016 and 039.329/2016); (c) Vicerrectoría de Investigación Universidad de Concepción (DIUC 208.603.009-1.0); (d) Programa de Mitigación de Riesgos, Ministerio del Interior y Seguridad Pública "Proyecto de manejo integrado de cuencas abastecedoras de agua en la Provincia de Malleco"; and (e) the project "Ecosystem-based strategies and innovations in water governance networks for adaptation to climate change in Latin American Landscapes" (EcoAdapt) funded

by the European Commission under FP7 contract ENV.2011.4.2.3-1/283163. Special gratitude is extended to the Chilean Forest Service (Corporación Nacional Forestal: CONAF) for helping with fieldwork and giving us the support to develop dendrochronological research in Araucaria ecosystems. We also acknowledge Nick Crane for his help in an earlier version of this manuscript.

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

560

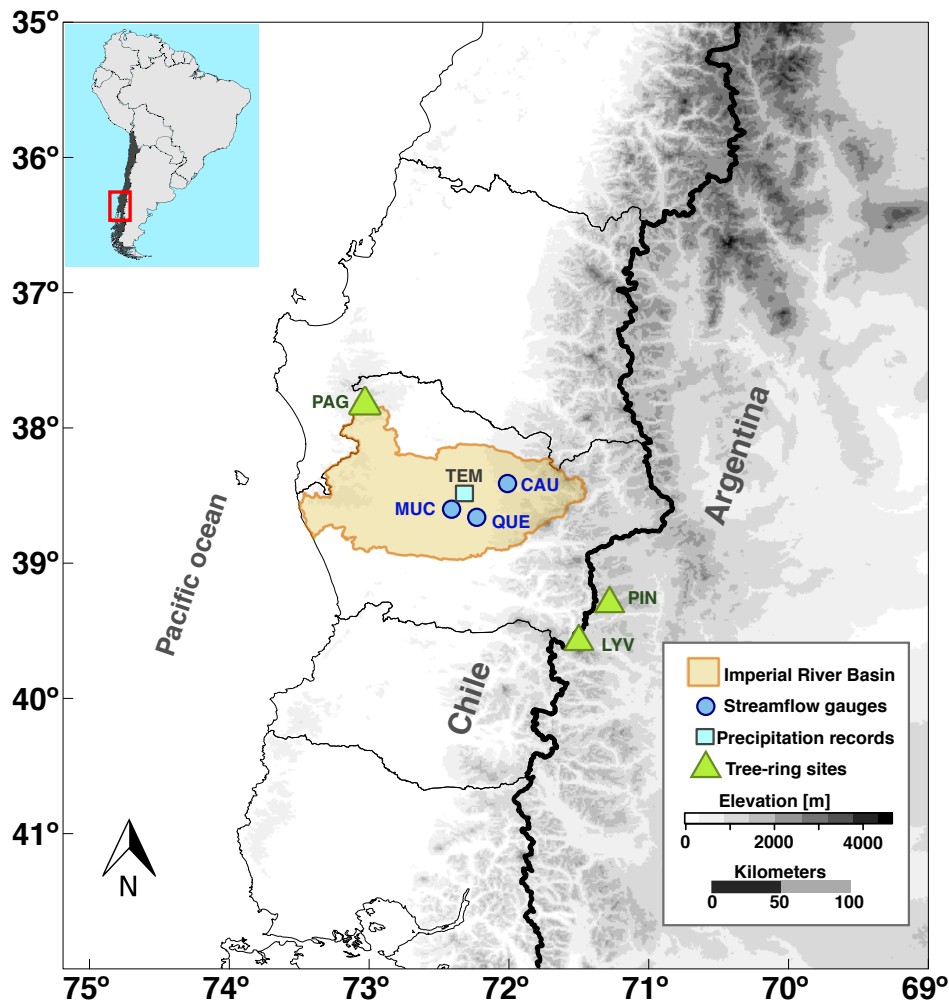

**Figure 1.** Map showing the Río Imperial hydrological basin in the context of the SCC region. Locations of instrumental records and tree-ring sampling sites are also included

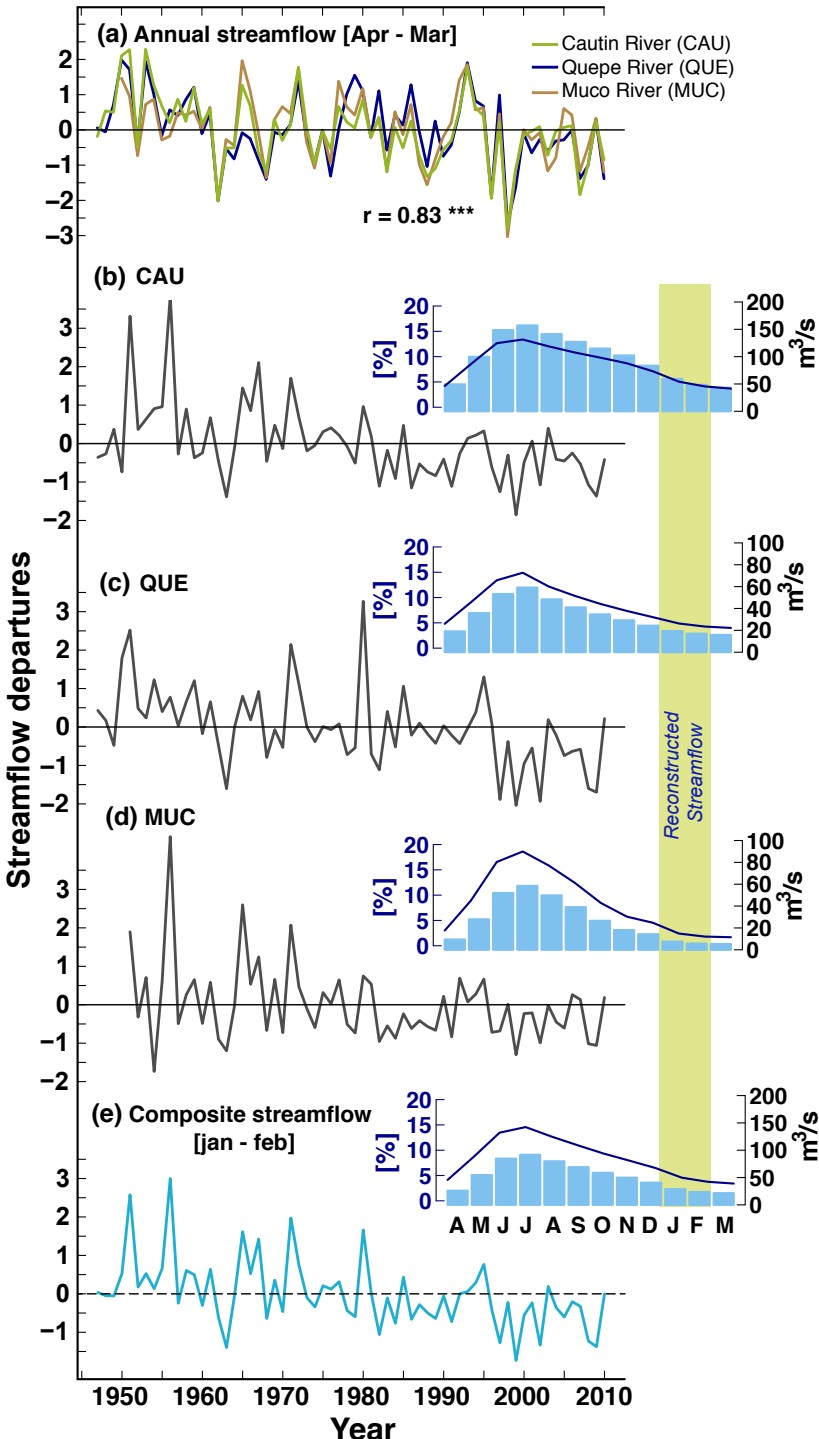

**Figure 2.** Hydrological regime of the instrumental records utilized in this study. These records were combined to develop a composite summer streamflow record for the period 1947-2010. The plots showing "streamflow departures" correspond to standardized anomalies for the common period of the records. The hydrographs depict the average streamflow of each month (bars) and the proportion of yearly discharge each month contributes (curves).

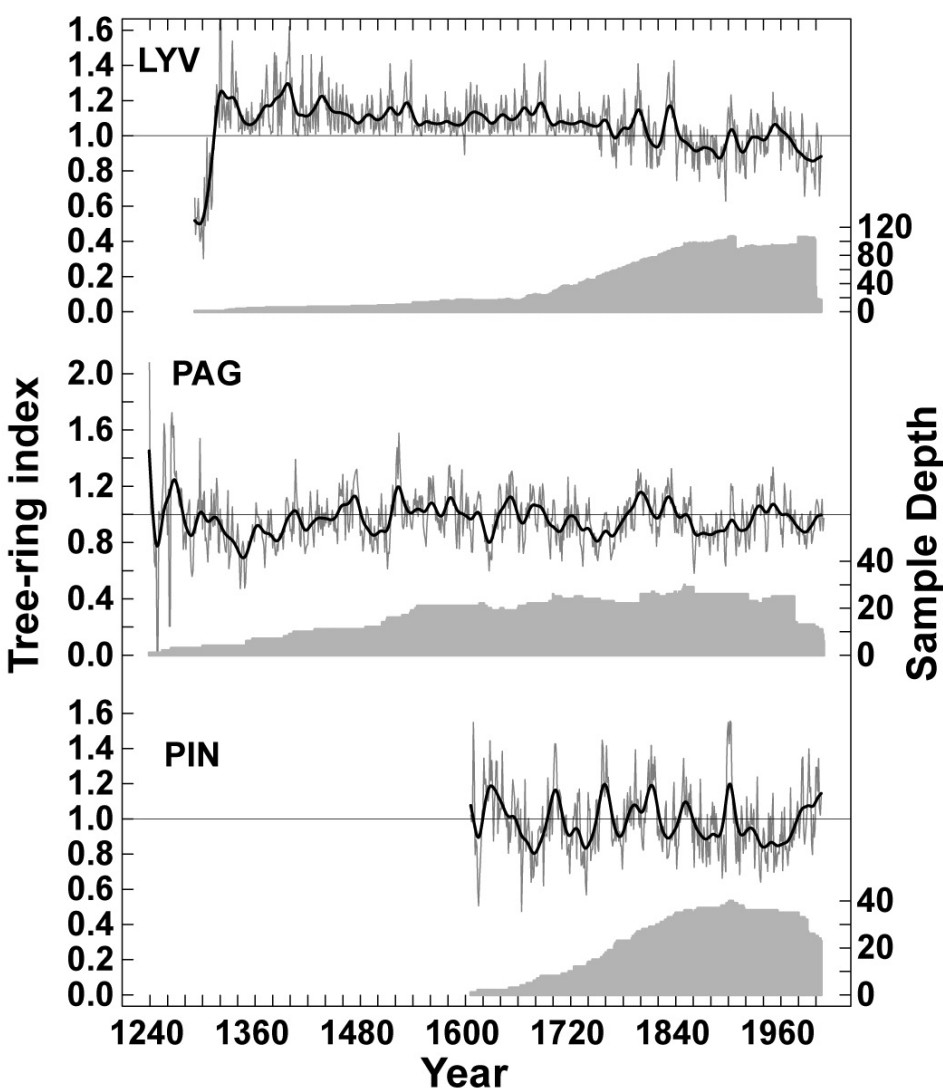

**Figure 3.** *Araucaria araucana* tree-ring chronologies utilized for the Río Imperial streamflow reconstruction. Gray areas represent the number of tree-growth series included in each chronology. The longest time series corresponds to the one sampled in the Nahuelbuta National Park (PAG), representing the period 1239-2009.

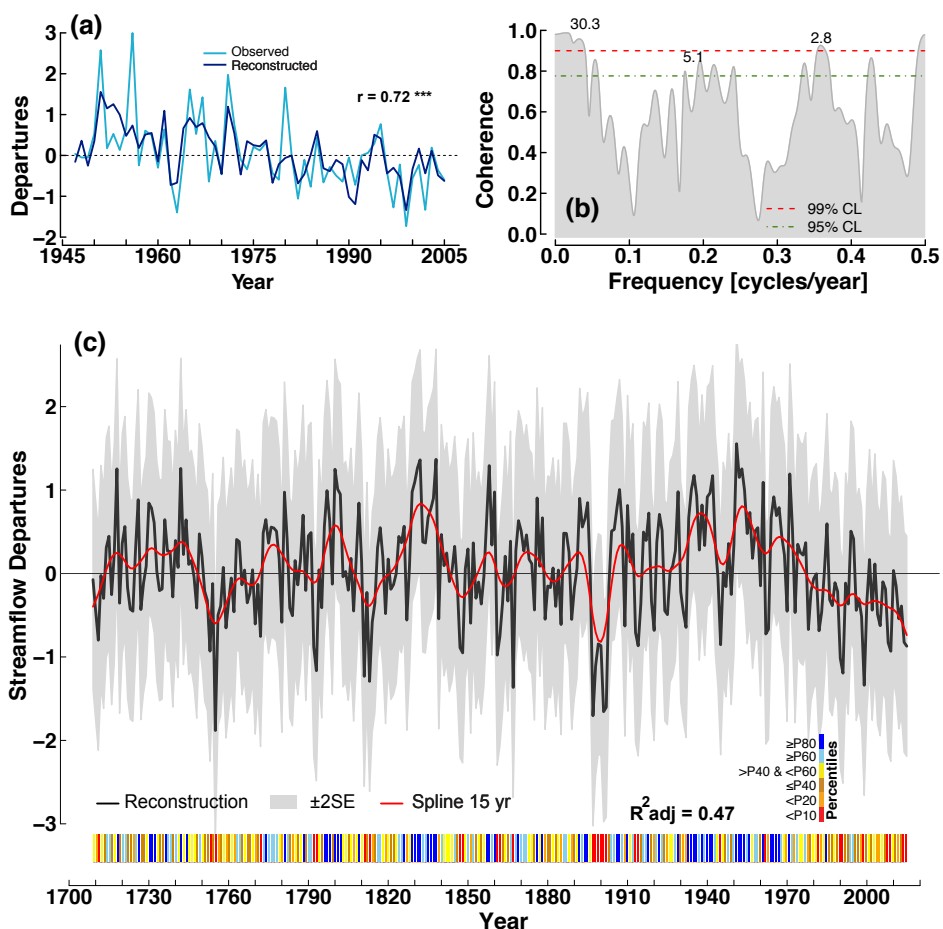

**Figure 4.** Summer streamflow reconstruction for the Río Imperial. The top-left panel compares standardized anomalies of observed (instrumental) streamflow and our tree-ring reconstruction. The top-right panel depicts the coherence of cycles between both time series. The lowest panel corresponds to the streamflow reconstruction, including the envelope for the $\pm2$ standard error (SE), a spline filter, and a classification of each year in the percentiles of the probability distribution (bottom section).

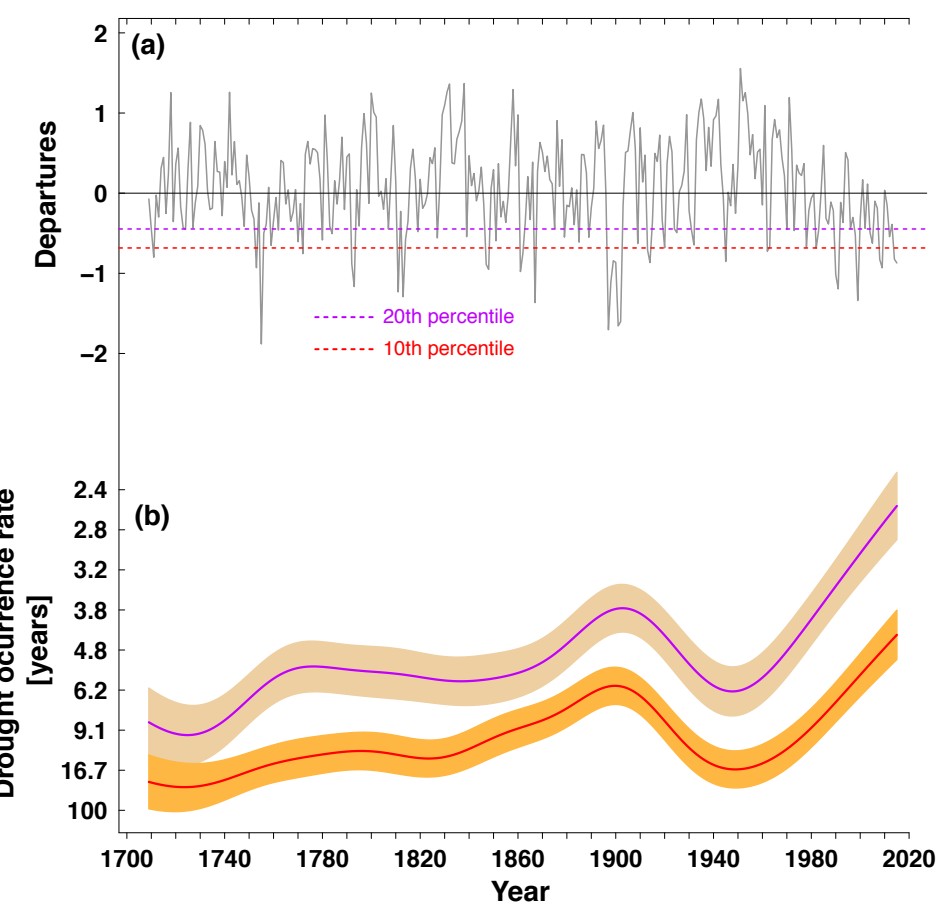

**Figure 5.** Summer streamflow reconstruction and the periods of low flow. In (a) the 10[th] (red) and 20[th] (purple) percentiles are plotted over the time series. (b) shows the return period of each low flow (below the 10[th] and 20[th] percentiles) for the whole record.

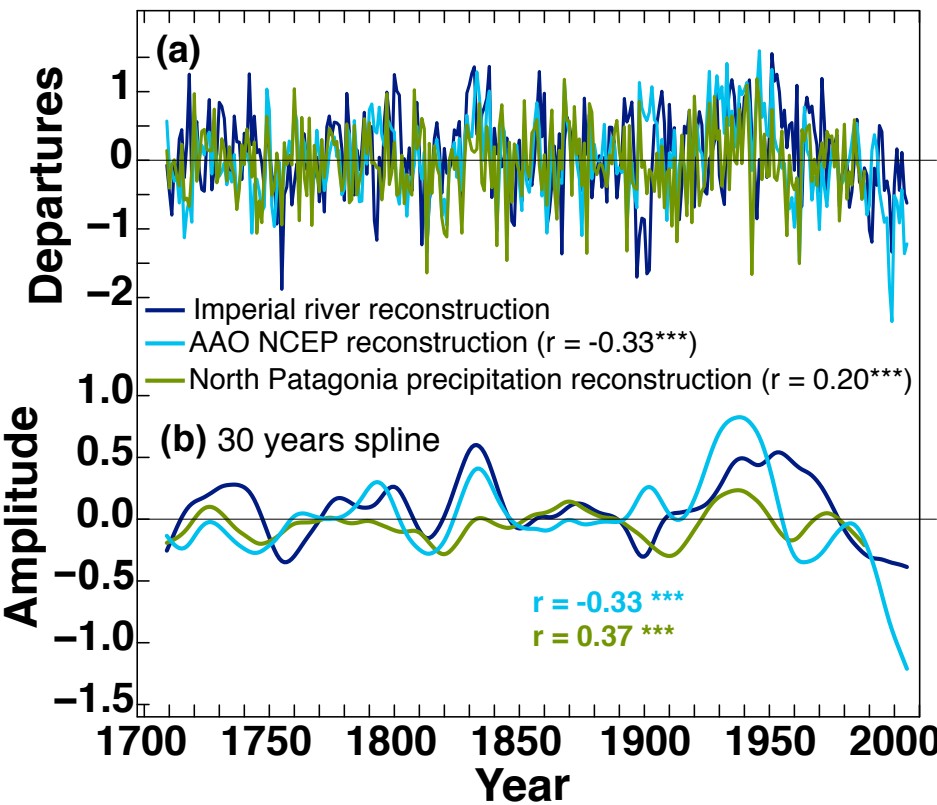

**Figure 6.** Comparison between the tree-ring reconstruction of Río Imperial streamflow *versus* the tree-ring precipitation reconstruction presented in Villalba et al. (1998) and the AAO tree-ring reconstruction from Villalba et al. (2012) that also utilized an index of SAM based on pressure fields from NCAR-NCEP reanalysis (Thompson and Wallace, 2000). In (a) these time series are displayed at annual resolution while (b) shows the same time series after the application of a 30-year spline filter. In both panels, the AAO index has been inverted to facilitate comparison with respect tree-ring reconstructions. Pearson correlations between the streamflow reconstruction and the other time series are included at the bottom of each panel, with asterisks representing statistically significant correlations at p-value < 0.001.

**Table 1.** Instrumental streamflow records utilized in this study. These records were retrieved from the Chilean National Water Authority (DGA or Dirección General de Aguas).

| Name | Latitude (S) | Longitude (W) | Period | Altitude (m) | Discharge $(m^3s^{-1}a^{-1})$ | Coefficient of Variation |
|---|---|---|---|---|---|---|
| Cautín at Rari-Ruca | 38°55' | 72°00' | 1943-2010 | 425 | 98.1 | 16.43 |
| Quepe at Vilcún | 38°41' | 72°13' | 1946-2010 | 292 | 33.1 | 13.57 |
| Muco at puente Muco | 38°37' | 72°25' | 1951-2010 | 250 | 26.2 | 37.88 |

**Table 2.** Rank of the five years with highest and lowest discharges as calculated from the composite built from the instrumental record (1947-2010). Departures are expressed as percentages relative to the mean (i.e. $\%=Year_x/\bar{x}$, where $Year_x$ corresponds to the streamflow of any given year and $\bar{x}$ to the mean of the period).

| Ranking | High flow (% of the mean) | Low flow (% of the mean) |
|---|---|---|
| 1 | 1956 (209.8) | 1999 (43.8) |
| 2 | 1951 (184.1) | 1963 (54.0) |
| 3 | 1971 (166.8) | 2009 (55.4) |
| 4 | 1965 (159.8) | 1997 (57.2) |
| 5 | 1980 (150.1) | 2008 (59.9) |

**Table 3.** Main features of the tree-ring chronologies developed for this study. The Lanín (LAN) and Villarrica (VILL) chronologies correspond to those presented in Mundo et al. (2012) and González et al. (2005), respectively.

| Location | Code | Latitude (S) | Longitude (W) | Period | Altitude (m) | Source |
|---|---|---|---|---|---|---|
| Pinalada Redonda | PIN | 39°18' | 71°17' | 1606-2006 | 1119 | Mundo et al. (2012) |
| Piedra del Águila | PAG | 37°50' | 73°02' | 1239-2009 | 1300 | Muñoz et al. (2014) |
| Lanín y Villarrica | LYV | 39°35' | 71°30' | 1291-2006 | 1350 | Composite LAN+VILL |

**Table 4.** Statistical description of the tree-ring chronologies developed for this study. The column labelled as "EPS" depicts the year when that statistic began to be larger than 0.85.

| Code | EPS | Autocorrelation | Average sensitivity | VIF |
|------|------|-----------------|---------------------|-------|
| PIN | 1750 | 0.591 | 0.230 | 1.011 |
| PAG | 1450 | 0.490 | 0.170 | 1.739 |
| LYV | 1525 | 0.566 | 0.206 | 1.724 |

**Table 5.** Rank of the five periods showing highest and lowest discharges as calculated from the tree-ring reconstruction. These periods were organized according to windows of five, 10, and 20 years. At the bottom of each period we included the highest/lowest streamflow from the instrumental record. Periods in bold represent coincidence between instrumental and reconstructed streamflow.

| Period length (years) | Rank | High flow | Low flow |
|---|---|---|---|
| | 1 | 1951 | 1755 |
| | 2 | 1838 | 1897 |
| 1 | 3 | 1832 | 1901 |
| | 4 | 1858 | 1902 |
| | 5 | 1940 | 1867 |
| | *Instrumental record* | 1956 | 1999 |
| | 1 | 1951-1955 | 1897-1901 |
| | 2 | 1829-1833 | 1753-1757 |
| 5 | 3 | 1938-1942 | 1811-1815 |
| | 4 | 1797-1801 | 1996-2000 |
| | 5 | 1935-1939 | 1987-1991 |
| | *Instrumental record* | 1955-1959 | 2011-2015 |
| | 1 | 1829-1838 | 1896-1905 |
| | 2 | 1933-1942 | 1751-1760 |
| 10 | 3 | **1950-1959** | **1990-1999** |
| | 4 | 1796-1805 | 1982-1991 |
| | 5 | 1964-1973 | 1810-1819 |
| | *Instrumental record* | **1950-1959** | **1990-1999** |
| | 1 | 1934-1953 | **1986-2005** |
| | 2 | 1823-1842 | 1753-1772 |
| 20 | 3 | **1953-1972** | 1883-1902 |
| | 4 | 1728-1744 | 1803-1865 |
| | 5 | 1785-1804 | 1843-1865 |
| | *Instrumental record* | **1953-1972** | **1986-2005** |

**Table 6.** Comparison between streamflow observations and streamflow reconstructed versus ENSO and SAM for the period 1948-2005, where "r" corresponds to the Pearson correlation coefficient

| Streamflow record | Index | Months | r |
|---|---|---|---|
| Instrumental | AAO | Oct-Sep | -0.58 |
| Instrumental | SOI | Feb-Mar | 0.32 |
| Instrumental | AAO | Sep-Dec | -0.54 |
| Tree-ring | AAO | Sep-Aug | -0.66 |
| Tree-ring | AAO | Dec-Feb | -0.50 |
| Tree-ring | SOI | Mar | 0.36 |
| Tree-ring | AAO | Sep-Dec | -0.54 |