# Peer review of "Dendrohydrology and water resources management in South-Central Chile: Lessons from the Río Imperial streamflow reconstruction"

_Hydrology and Earth System Sciences, 2017_

## Referee Comment (RC1) · Anonymous Referee #1 · 24 Oct 2017

I found the study very interesting. The reconstruction of river flows is important by itself, providing a long historical record that improve statistics. Similar mega-droughts in the past can provide insights about drivers when these findings are compare with similar behaviours in other regions. I mean, the water management importance can be true, but is not fundamental for the acceptance of this work. According to my experience, the manuscript should be accepted with minor revisions.

Major comments

- Is the Rio Imperial streamflow variability representative of the entire SCC? I think it is not. Please provide evidences about this point, especially for summer. It is an

important issue due the title of this paper include a large region from 35°S to 42°S in Chile.

- How an accurate calculation of natural streamflow variability can help to anticipate possible consequences in the water management? This is the justification or motivation of the analysis. My point is the following. If you proved that streamflow variance (or extremes) in the past was larger than the expected by models for the future, how this information is useful for water management? Perhaps, if droughts were more severe in the past, without a major extinction or decrease of vegetation, you can ask why we expect a major problem in the future? The major uncertainty is related to in any case with the water demand, but not with the natural or anthropogenic origin of the droughts. It is the minimum ecological discharge the variable benefited by this study? If so, I guess the water management issue is restricted to the decisions depending on this variable. Can you focus on this point?

- It would have being very instructive if you were count with data until 2014 or 2015 with the aim of calculating the return period of droughts or the recurrence rate of drought events of the mega-drought mentioned in the work of Garreaud (2015). According to this author, the mentioned mega-drought 2010-14 is the largest on record. Can these statistics used in your work shows the extreme nature of the mega-drought? (at least for the instrumental record).

- I understand the use of the Southern Oscillation Index (SOI) instead the Niño 3.4 SST anomalies, due the longest record of atmospheric pressure data in Tahiti and Darwin. In fact, SOI is not usually used in climate studies, due its large "noise" in the intraseasonal timescale, compared to the more smoothed evolution of the SST index in the central equatorial Pacific. Anyway, I expected the used of the SOI directly calculated from stations but you used the NCEP-NCAR reanalysis. What is the justification then?

- About equation (1) and Fig. 3. I am not dendronologist, so it is surprising the low covariance shared between these reconstructions, even when the samples are larger.

The trees are not responding in the same way to the atmospheric forcing? (water availability for instance). This explain equation (1)... but still I need an explanation in the text for this behaviour. Looking at figure 4 (upper-left panel) the reconstruction looks good.

- Fig. 4. It is clear from the comparison between reconstructed and observed streamflow that the reconstruction captures the low frequency but not the interannual variance, although the coherence shows a peak on 2.8 years. Based on this finding, why the authors can expect a reliable comparison with ENSO? Because it is the most important driver at interannual timescales? In fact, what is the reason for not using the Interdecadal Pacific Oscillation instead ENSO? Why SAM? Please provide references that reinforce your thoughts about possible drivers.

- Page 7, lines 25-26. The sentence is misleading, because Garreaud (2015) and Boisier et al (2016) define the mega-drought since 2010, exactly when your information stop. Have you considered the possibility of interdecadal variability? IPO changes to its positive (warm) phase at the end of 70s, changing to a negative (cold) phase ant the end of 90s. This can be seen as a negative trend since 1980... Please, provide some discussion about this possibility.

On the other hand, the positive trend of SAM can be related to any trend, even without a physical explanation. In your results, when you remove the linear trend the correlation fall to near-zero values, so what is the reason there is a relationship between SAM and streamflow at 38°S? SAM it is just a long-term trend? Why there is not relation at other timescale? Do you know what is controlling the SAM trend? That is a key answer to make.

I think you should read the following paper: http://www.scielo.org.mx/pdf/atm/v25n1/v25n1a1.pdf

Minor comments

- Page 2, lines 10-11. This values were taken from the figures? If so, how accurate is

that?

- Page 2, lines 12-15. You have written 3 times "in this region" in few lines. I think it can be improved.

- Page 4, line 17. Where is mentioned Table 2?

- Table 2. What instrumental streamflow record have you used?

- Page 4, line 15. I am not expert on dendronology, so I can not questioning the methodologies employed to construct the index based on three tree-ring chronologies. About equation (1), however, there is something intriguing to me. I assume that water availability affects in the same way same species of trees. Why coefficients are opposite in sign for PAG (+2.69) and LYV (-1.97) at the same time (t-1)?

- Page 4, line 24. It is written "the return period or extreme low flows..." Did you mean "the return period of extreme low flows"?

- Page 5, line 21. You defined summer ad January-February for streamflow. So, what are the previous months for rainfall and what is the value for the not simultaneous correlation"?

- Table 2 is analysed in page 5. I suggest to exchange numbers with table 3.

- Page 5, lines 26-27. Clearly, streamflow as precipitation exhibits a positive skewness in southern Chile, which is a normal behaviour taking into account that at most, there will be no rainfall (0 mm) as the lowest values. This kind of distributions are typical also for wind speed. So, I do not understand the point of this sentence.

- Page 6, line 8. Define VIF.

- Fig. 4. It is very nice the percentiles at the bottom of the figure. Easy to interpret. I wonder what would be the percentile for the period 2010-2015? This information is available, why you do not have used?

[Figure]

- Page 8, lines 33-34. The summer of 1999 is part of La Niña, not El Niño. In fact, the winter of 1998 is one of the most dry winters in instrumental record.

- Page 10, lines 24-25. The restriction of power supply occurred in 1996, at least in Santiago. It was different in Temuco?

---

## Referee Comment (RC2) · Anonymous Referee #2 · 29 Nov 2017

This manuscript is a case study for reconstruction of streamflow based on tree-ring growth data in Chile. Just to put things in perspective, and I do not mean any negativity here, there is nothing new in this particular manuscript with regard to dendrohydrology. There is almost 50 years of literature in this area; the same regression models, the same way of reconstruction, and the same variable (streamflow). The authors reconstructed summer flows, instead of annual flows, but reconstruction of seasonal variables rather than annual ones happened also several times before but perhaps not in Chile. So, from my point of view, there is not any aspect of novelty in this research. It is important to state this, at least to give the authors a chance to clarify in case I missed something, and I apologize if I did. The section that is most interesting to read

is section 4, which the authors called "Discussion and conclusion". In this section, the authors argued hard for the utility of tree ring-based reconstruction to identify droughts that are more sever or more frequent than those inferred from the instrumental record. This argument can be found in so many of the dendrohydrology papers published in the last decade, and again nothing here is new. However, I have a fundamental issue with the scientific foundation of the argument, and unfortunately this applies also to several other papers published in this field. The authors reconstruct almost 300 years of streamflow data and start comparing it with instrumental record of 60 years. Obviously, they find droughts in the 300 years with characteristics that are different from those in the 60 years, of course! But for water resources engineers, whatever you find in 300-year record MIGHT be a 300-year drought. It is unfair to compare it with the 60-year record. Engineers would fit a statistical distribution to the 60-year record, estimate 100 or 1000-year droughts, then fit a distribution to the longer reconstructed record, and again estimate whatever drought quantiles you want, then compare. Otherwise, engineers and water resources planners never use just the deterministic short instrumental record of flows. If you can prove, based on the analysis I suggested, that the reconstructed flows lead to significantly different frequency or severity of droughts, then you made the case about the utility of reconstruction. Other specific points: Page 2, Lines 8-10: If you have the future projections based on the CMIP5 results, why don't you investigate if projected future droughts are more sever/frequent than the past ones? Just an idea; Page 3, Line 27: How did you get the "natural regime"? Did you account for irrigation abstraction? Page 3, Line 29: This sentence is not clear. How did you use the double mass curves to determine the calibration window of time? Page 4: I feel that I miss proper information about the hydroclimatology of the region. How wet or dry is it? How much is the rain and its variation? Just provide some background information; Page 4, Line 17: You are referring to Table 3 before Table 2, please reorder the Tables; Page 4, Lines 24-26: peak over threshold is usually used for floods, but here you are doing drought analysis. Do you mean flow below threshold or something like this? You also need a reference for portion; Page 5, Line 25: What are these percentage numbers (54.06% and 74.12%)? Do you mean m3/s? Page 6, Line 2: "that" should be "than", and "here" is not clear. Do you mean your manuscript? If so, why is the reference, it is confusing; Page 6, Lines 11-17: These results are not really good (especially, RE of 0.36), I know they are typical in many dendrohydrology studies, but they should at least make the authors a bit more humble and lighten the assertion tone that is coming later in Section 4; Page 6, Lines 23-24: What does this sentence mean? Page 6, Lines 25-27: Is there any meaning for these windows of 5-year, 20-year,..etc? Of course, every time you change the window, you can get different results, but what are we supposed to learn from this? Page 6, Line 27: You cannot really use reconstructed flows to comment on extremely high streamflow. Look at your Figure 4 (top left) and you will agree with me; Page 6, Lines 29-33: I cannot understand this portion; Page 7, Lines 1-5: What does this argument imply? Rain and streamflow are different! So, how did you conclude that it is a pluvial system? I think you need to elaborate; Pages 7-11: Almost half of the paper came under one section called Discussion and Conclusions. This is a style and format issue that does not look good. You need to include more analysis with the Results section, then not very long Discussion section, then a separate Conclusions section, this will be better. Page 7, Line 31: Usually trend analysis is misleading. Have you looked at the trend of the entire reconstructed record? Page 8, Line 5: I got confused, was that SAM work done in this study or taken from other studies? Page 8, Lines 10-12: On what basis was this statement made? Looking at Figure 6 and the correlation numbers does not give me the same impression that the authors have; Page 11, Lines 21-24: I cannot find proof for this in the manuscript, perhaps the authors need to rewrite this; Figure 2 (but also a general comment): On what basis was the selection of Jan-Feb only? Why not March and April too? They also seem part of the low flow season, especially that it may not be a good idea to call two months a drought; The authors also need to note that averaging the streamflow of three stations may reduce the variability in individual gauges, and make the reconstruction easier (nevertheless, the reconstruction accuracy is not high anyways). So, you need to justify this; Table 1: Please report the standard deviation also or better,

the coefficient of variation to see the variability of each series; Table 3: What are those LAN and VILL in the Table title? Table 4: The autocorrelation of the tree ring chronologies are quite high, and is usually transferred to the reconstructed annual flows. I find this unrealistically high for annual flows, can you comment on this and its impact on the reliability of the reconstructed flows? Can you compare it with the autocorrelation of the instrumental flows?

---

## Author Comment (AC1) · 26 Jan 2018

**Responses to Reviewer 1**

**We are very grateful with the comments provided by the anonymous Reviewer 1. It is clear that her/his comments come from a deep analysis of the work we present here. Below you will find our response to each.**

Is the Rio Imperial streamflow variability representative of the entire SCC? I think it is not. Please provide evidences about this point, especially for summer. It is an important issue due the title of this paper include a large region from 35◦S to 42◦S in Chile.
**RESPONSE: Thanks to your observation we reassessed the sentences including the 35-42 strip and decided to modify the manuscript to "SCC ~37°-42°S". Previous studies (Rubio-Álvarez & McPhee, 2010; Muñoz et al., 2016; Lara et al, 2008) found that the Río Imperial fits better as representing the region south of 37.5, as for example a recent reconstruction of the BioBío river (~37.5°S) was similar to another built for the Puelo river (~42°S). However, we consider important to indicate that available observational datasets for the period 1980-2010 suggest that the region south of 35-42°S is a hydroclimatic cluster (see González-Reyes et al 2017), especially during the summer, our target season. The figure R1.A shows that our reconstructed streamflow (dark blue) correlates significantly (as described by the p-value) with observed streamflow of each individual station, and also each individual station correlates significantly with the composite time-series produced in our study. We will include this new analysis in the paper in order to provide further evidence on the relevance of the Río Imperial reconstruction for this region.**

How an accurate calculation of natural streamflow variability can help to anticipate possible consequences in the water management? This is the justification or motivation of the analysis. My point is the following. If you proved that streamflow variance (or extremes) in the past was larger than the expected by models for the future, how this information is useful for water management? Perhaps, if droughts were more severe in the past, without a major extinction or decrease of vegetation, you can ask why we expect a major problem in the future? The major uncertainty is related to in any case with the water demand, but not with the natural or anthropogenic origin of the droughts. It is the minimum ecological discharge the variable benefited by this study? If so, I guess the water management issue is restricted to the decisions depending on this variable. Can you focus on this point?
**RESPONSE: We could not agree more with this assessment; the major uncertainty is related to water demand, which we don't analyze in detail here. However, we want to clarify that the main result in our work is the increasing frequency of dry summers, which is unseen in the analyzed period. In our view, this implies a new hydroclimatic scenario that might make the region more sensitive to changes in water demand, such as a possible utilization of all water rights in the watershed. In the abstract, results and discussion section we have modified the text in order to emphasize these findings. On the other hand, that our work finds some summers comparatively drier than what is seen in the available observations show, suggests that calculations of return periods can benefit from research like ours in order to produce fine-tuned statistics, specially in the determination of confidence intervals for the minimum ecological discharge. This is a statement we already have in our paper but it may be clear if we divide the section of discussion and conclusion in two, as the reviewer number 2 suggests.**

[Figure]

**FIGURE R1.A: Comparison of available streamflow data (red), our tree-ring reconstruction (dark blue), and the composite streamflow for the period 1980-2010 (cyan).**

It would have being very instructive if you were count with data until 2014 or 2015 with the aim of calculating the return period of droughts or the recurrence rate of drought events of the mega-drought mentioned in the work of Garreaud (2015). According to this author, the mentioned mega-drought 2010-14 is the largest on record. Can these statistics used in your work shows the extreme nature of the mega-drought? (at least for the instrumental record).

**RESPONSE: We thank you very much for your suggestion. We have decided that comparing our results with the recent developments as depicted in Garreaud et al (2015) is important and necessary. We extended our reconstruction including the period 2011-2015 ("megadrought") and found that, despite being an unprecedented dry period for the instrumental record, it does not rank as the highest in the reconstruction (see Table R1.A below). We think this finding is very interesting because a recent paper studying the megadrought on winter-spring precipitation found that the period 2011-2015 as highly unusual in the last 1000 years (Garreaud et al 2017). We also find this period as the driest of the streamflow instrumental record, but it is not too different from the period 1996-2000. In addition, in the reconstruction the 2011-2015 period ranks fifth but with an anomaly that is less than a half of the driest (1897-1901). The Figure R1.B shows that the summer reconstruction for 2011-2015 is far from the most extreme, but corroborates our main finding that dry years become more recurrent since 1980. We will update our results and discussion with these findings.**

**Table R1.A: Updated 5-year rankings for the instrumental and reconstructed periods**

| Reconstructed period (1709-2015) | Low flow | | High flow | |
|---|---|---|---|---|
| | 5-yrs | reco | 5-yrs | reco |
| | 1897-1901 | -1.235 | 1951-1955 | 1.087 |
| | 1753-1757 | -0.764 | 1829-1833 | 1.016 |
| | 1811-1815 | -0.737 | 1934-1938 | 0.842 |
| | 1996-2000 | -0.586 | 1834-1838 | 0.816 |
| | 1987-1991 | -0.584 | 1797-1801 | 0.751 |
| | **2011-2015** | **-0.556** | 1939-1943 | 0.749 |

| Instrumental period (1947-2015) | | | | |
|---|---|---|---|---|
| | 5-yrs | reco | 5-yrs | reco |
| | **2011-2015** | **-0.778** | 1955-1959 | 0.972 |
| | 1996-2000 | -0.777 | 1950-1954 | 0.841 |
| | 2005-2009 | -0.681 | 1965-1969 | 0.727 |
| | 2008-2012 | -0.677 | 1971-1975 | 0.560 |
| | 1987-1991 | -0.382 | 1976-1980 | 0.271 |
| | 1982-1986 | -0.378 | 1992-1996 | 0.200 |

[Figure]

**Figure R1.B: Updated calculations of departures and recurrence intervals for the period studied, including the megadrought (2011-2015).**

I understand the use of the Southern Oscillation Index (SOI) instead the Niño 3.4 SST anomalies, due the longest record of atmospheric pressure data in Tahiti and Darwin. In fact, SOI is not usually used in climate studies, due its large "noise" in the intraseasonal timescale, compared to the more smoothed evolution of the SST index in the

central equatorial Pacific. Anyway, I expected the used of the SOI directly calculated from stations but you used the NCEP-NCAR reanalysis. What is the justification then?

**RESPONSE: Thank you very much for catching this up. We will re-write these sentences since they don't make it clear we do use a SOI index built from observational data. We utilized the SOI index that begins in 1866 and is downloaded from the following link: http://www.cgd.ucar.edu/cas/catalog/climind/soi.html.**

About equation (1) and Fig. 3. I am not dendronologist, so it is surprising the low covariance shared between these reconstructions, even when the samples are larger. The trees are not responding in the same way to the atmospheric forcing? (water availability for instance). This explain equation (1)... but still I need an explanation in the text for this behaviour. Looking at figure 4 (upper-left panel) the reconstruction looks good.

**RESPONSE: We appreciate this comment that allows us to further clarify our analysis of individual chronologies. LYV and PAG correlate positively with same-year streamflow (0.38 and 0.43, respectively), with the streamflow of the previous year (0.53 and 0.41, respectively), and with streamflow two years before (0.3 and 0.23, respectively). Conversely, PIN shows negative correlations with previous years (-0.56 for the 1$^{st}$ and -0.38 for the 2$^{nd}$ previous year). The negative correlation for PIN is because the site is a narrow sector with recurrent fog, where atmospheric moisture and precipitation in summer may produce a relative reduction in incoming radiation and temperature; these conditions may reduce the rate of the tree-ring growth. Mundo et al (2012) already found this behavior for PIN as part of a large group of chronologies. We will include this explanation in our manuscript.**

Fig. 4. It is clear from the comparison between reconstructed and observed streamflow that the reconstruction captures the low frequency but not the interannual variance, although the coherence shows a peak on 2.8 years. Based on this finding, why the authors can expect a reliable comparison with ENSO? Because it is the most important driver at interannual timescales? In fact, what is the reason for not using the Inter-decadal Pacific Oscillation instead ENSO? Why SAM? Please provide references that reinforce your thoughts about possible drivers.

**Response: We fully agree with the reviewer in that the reconstruction fits well with the low frequency, but we also believe it does well with the interannual variability because there are only two peaks not being captured. Figure 1 shows no peak on these years for Quepe (QUE ~1956) and Muco (MUC, ~1982) but these features are averaged-out in the composite. Thus, although our reconstruction does not capture every fluctuation of the composite time-series, it likewise evidences correspondence with observations. Nevertheless, we applied new analyses to the data. Utilizing the Blackman-Tukey spectral method (Ghil et al 2002, Figure R1.C) we see that the Río Imperial reconstruction has high frequency cycles (2-7 years) and mid-to-low frequency (> 8 years). A Multi-taper method and a Singular Spectral Analysis reveal that a ~4-year cycle captures 10% of the variance while a ~7-year cycle captures 7%. On the other hand, a 16-20-years cycle corresponds to 20%. We also performed a Continuous Wavelet Transform Analysis and found that a 16-32-years frequency is significant between 1800-1950; high frequency cycles occur along the whole period but appear more significant after 1900.**

[Figure]

**Figure R1.C: Spectral Analyses of the time-series demonstrating that it captures high frequency**

Regarding ENSO, there is abundant literature correlating it with hydrological variables at the annual scale, but there is no such complete information for the summer. As we explain in our manuscript, Urrutia et al (2011) find annual discharge in the Maule river (~35°S) well correlated with ENSO, while Muñoz et al

**(2016) find the BioBío river (~37°S) correlated with SAM. This was the criteria we utilized to perform these correlations. The río Imperial is south from the BioBío and our work is the first solely focused on summer dynamics. Barría et al (2017) analyzed the upper section of the BioBío dividing the year in two sections: October-March (OM) and April-September (AS). They find that PDO correlates negatively with AS runoff and positively with OM; For SAM, they find positive correlation for OM. Giving your comment and these new studies, we found no reason to not perform comparison between our time series and the IPO. We compared our time-series with the IPO reconstruction presented in Vance et al (2015) and found a statistically significant negative correlation between IPO and the 16-20-year cycle of our reconstruction (-0.38, n=295), although it looks more coherent during the 20[th] century (Figure R1.D). We will include these new analyses into the manuscript.**

[Figure]

 **Figure R1.D: Comparison between our reconstruction and the IPO (curve inverted for readability).**

Page 7, lines 25-26. The sentence is misleading, because Garreaud (2015) and Boisier et al (2016) define the mega-drought since 2010, exactly when your information stop. Have you considered the possibility of interdecadal variability? IPO changes to its positive (warm) phase at the end of 70s, changing to a negative (cold) phase ant the end of 90s. This can be seen as a negative trend since 1980... Please, provide some discussion about this possibility.

**Response: We agree with this comment. We reassessed that sentence and find it misleading. In our work we find that dry years are more recurrent since ~1980 and in the new version of the manuscript we highlight that this trend is previous to the mega-drought identified in Garreaud (2015) and Boisier et al (2016).**

On the other hand, the positive trend of SAM can be related to any trend, even without a physical explanation. In your results, when you remove the linear trend the correlation fall to near-zero values, so what is the reason there is a relationship between SAM and streamflow at 38∘S? SAM it is just a long-term trend? Why there is not relation at other timescale? Do you know what is controlling the SAM trend? That is a key answer to make. I think you should read the following paper:

http://www.scielo.org.mx/pdf/atm/v25n1/v25n1a1.pdf

**RESPONSE: We do not completely agree with this assessment. First, we believe we provide a mechanistic explanation of the influence of SAM on precipitation in the region (Page 8 L.5 to Page 9 L.5). Nevertheless, we see that the recommended paper is very helpful for us in order to improve this explanation. Second, the relation between SAM and our data is not near zero; we performed correlations using a prewhitened version of the reconstruction and observations (removing the lag-1 autocorrelation) and they are negative and statistically significant (-0.287 with p-value=0.033 and -0.290 with p-value=0.029, respectively).**

Minor comments

Page 2, lines 10-11. This values were taken from the figures? If so, how accurate is that?
**RESPONSE: Yes, the data had been read from the figures. We reassessed that section and decided to utilize the information provided by the Atlas of Global and Regional Climate Projections of the IPCC (IPCC, 2013). We eliminated the sentence in Page 2-Lines 7-13 and replaced it with the following: "SCC is expected to undergo important climate changes. Analysis of the multi-model ensemble for the scenario RCP4.5 presented in the Atlas of Global and Regional Climate Projections (IPCC, 2013) indicates 10 to 30% reduction in spring and summer (October to March) precipitation by 2016-2035 and 2046-2065 relative to 1986-2005. The same projection forecasts 0.5 to 2°C warming for summer (December-February). Drier and warmer summers for may make SCC more vulnerable to water scarcity, given that this is the season of highest water demand in this region (Garreaud, 2015).**

Page 2, lines 12-15. You have written 3 times "in this region" in few lines. I think it can be improved.
**RESPONSE: Thanks for catching this up. We replaced the second "in this region" for "here" and the third for "SCC".**

Page 4, line 17. Where is mentioned Table 2?
**RESPONSE: Thanks for finding this omission. We have mentioned the table besides Fig. 1**

Table 2. What instrumental streamflow record have you used?
**RESPONSE: This is from the composite time-series. We have clarified this in the caption.**

Page 4, line 15. I am not expert on dendronology, so I can not questioning the methodologies employed to construct the index based on three tree-ring chronologies. About equation (1), however, there is something intriguing to me. I assume that water availability affects in the same way same species of trees. Why coefficients are opposite in sign for PAG (+2.69) and LYV (-1.97) at the same time (t-1)?
**RESPONSE: We appreciate this comment as it allows us to provide a deeper explanation of our procedures. In order to fully understand the reasons behind this kind of multi-regression model it is important to consider that the climate signal of a given ring-width is product of climatic conditions of the same year but**

**from previous years as well, and they can be different in certain moments and as results of different locations of the chronologies. In our study region, climatic conditions influence ring-growth in two main ways: (1) Current and previous year(s) snow accumulation on mountain areas can delay the ring's growing season, increasing the likelihood of negative correlations (and LYV is located at high elevation). (2) In high elevation sites temperature can be a limiting factor for ring-growth, this is because increase in temperature in this region is related to less moisture and rainfall, possibly producing negative correlations. Thus, PAG and LYV correspond to different landscape, one at high elevations (LYV) where some variable (e.g. temperature or seasonal snow) can explain certain portion of the correlation with hydroclimatic observations, and another at low elevations (PAG) where the relationship between ring-growth and soil moisture is more direct because for instance precipitation is always liquid. What it is important to keep in mind is that they are significantly correlated with the observational record and the statistical model is skilled in representing the streamflow variability. Muñoz et al (2016) and Lara et al (2015) are other two examples where the statistical model presents coefficients of inverse signal in the same year.**

Page 4, line 24. It is written "the return period or extreme low flows..." Did you mean "the return period of extreme low flows"?
**RESPONSE: Yes, "of" is correct. Changed**

Page 5, line 21. You defined summer ad January-February for streamflow. So, what are the previous months for rainfall and what is the value for the not simultaneous correlation"?
**RESPONSE: We find that your question gives us a good opportunity to further demonstrate the relevance of studying January-February streamflow. We ran correlations between observed/reconstructed streamflow with Temuco rainfall for each month of the previous year (Figure R1.E). These correlations are significant for December and February of the previous year (p-value <0.1 and <0.05, respectively) for the instrumental record. For the reconstruction, the correlation is significant for June and December (p-value <0.05).**

Table 2 is analysed in page 5. I suggest to exchange numbers with table 3.
**RESPONSE: We do not agree with this suggestion. Table 2 follows Table 1 in the sense the Table 1 presents the instrumental record and Table 2 provides the information for the analysis of that instrumental record. Then Table 3 appears because it is about the tree-ring chronologies. If we do the change suggested, we feel the manuscript loses readability and that our line of argument weakens.**

Page 5, lines 26-27. Clearly, streamflow as precipitation exhibits a positive skewness in southern Chile, which is a normal behaviour taking into account that at most, there will be no rainfall (0 mm) as the lowest values. This kind of distributions are typical also for wind speed. So, I do not understand the point of this sentence.
**RESPONSE: We believe this sentence is clear since it provides summary statistics corresponding to Table 2, which is about ranking streamflow extremes. We want to stress our analysis is about base-flow, which should rarely go to zero as rainfall and wind speed do.**

[Figure]

**Figure R1.E: Correlation of observed/reconstructed streamflow with Temuco rainfall for each month of the previous year.**

Page 6, line 8. Define VIF.
**RESPONSE: We will include the following definition of VIF in the method's section: The Variance Inflation Factor (VIF) evaluates the multicollinearity of the predictors; a VIF close to 1 means a low or no multicollinearity (Haan 2002) while a value above 10 is associated with multicollinearity problems between predictors (O'Brien 2007).**

Fig. 4. It is very nice the percentiles at the bottom of the figure. Easy to interpret. I wonder what would be the percentile for the period 2010-2015? This information is available, why you do not have used?
**RESPONSE: We updated that figure (Figure R1.B) in a previous response.**

Page 8, lines 33-34. The summer of 1999 is part of La Niña, not El Niño. In fact, the winter of 1998 is one of the most dry winters in instrumental record.
**RESPONSE: Thanks for catching this up. We are certain that winter 1998 was in fact part of a strong La Niña rather than el El Niño. We have modified the section "and the strong El Niño event of 1998" in the following: " and a strong La Niña event in 1998-1999".**

Page 10, lines 24-25. The restriction of power supply occurred in 1996, at least in Santiago. It was different in Temuco?
**RESPONSE: Our reference (Fischer and Galetovic, 2001) and the Decree 287 of 1999 (available at https://www.leychile.cl/Navegar?idNorma=137602&idParte=)**

**indicate that restrictions in energy supply were implanted across the whole Central Interconnected System in 1999, which includes Temuco.**

**New references**

Decree 287 of 1999 (available at https://www.leychile.cl/Navegar?idNorma=137602&idParte=)

Garreaud, R. D., Alvarez-Garreton, C., Barichivich, J., Boisier, J. P., Christie, D., Galleguillos, M., LeQuesne, C., McPhee, J., and Zambrano-Bigiarini, M.: The 2010–2015 megadrought in central Chile: impacts on regional hydroclimate and vegetation, Hydrol. Earth Syst. Sci., 21, 6307-6327, https://doi.org/10.5194/hess-21-6307-2017, 2017.

Ghil, M., M. R. Allen, M. D. Dettinger, K. Ide, D. Kondrashov, M. E. Mann, A. W. Robertson, A. Saunders, Y. Tian, F. Varadi, and P. Yiou, Advanced spectral methods for climatic time series, Rev. Geophys., 40(1), 1003, doi:doi:10.1029/2000RG000092, 2002.

González-Reyes, Á., J. McPhee, D.A. Christie, C. Le Quesne, P. Szejner, M.H. Masiokas, R. Villalba, A.A. Muñoz, and S. Crespo, 2017: Spatiotemporal Variations in Hydroclimate across the Mediterranean Andes (30°–37°S) since the Early Twentieth Century. J. Hydrometeor., 18, 1929–1942, https://doi.org/10.1175/JHM-D-16-0004.1

Haan CT (2002) Statistical Methods in Hydrology, 2nd ed. Ames: Iowa State University Press.

IPCC, 2013: Annex I: Atlas of Global and Regional Climate Projections [van Oldenborgh, G.J., M. Collins, J. Arblaster, J.H. Christensen, J. Marotzke, S.B. Power, M. Rummukainen and T. Zhou (eds.)]. In: Climate Change 2013: The Physical Sci- ence Basis. Contribution of Working Group I to the Fifth Assessment Report of the Intergovernmental Panel on Climate Change [Stocker, T.F., D. Qin, G.-K. Plattner, M. Tignor, S.K. Allen, J. Boschung, A. Nauels, Y. Xia, V. Bex and P.M. Midgley (eds.)]. Cambridge University Press, Cambridge, United Kingdom and New York, NY, USA.

O'Brien R (2007) A caution regarding rules of thumb for variance inflation factors. Quality & Quantity 41: 673–690.

Vance, T. R., J. L. Roberts, C. T. Plummer, A. S. Kiem, and T. D. van Ommen (2015), Interdecadal Pacific variability and eastern Australian mega-droughts over the last millennium, Geophys. Res. Lett., 42, 129–137, doi:10.1002/2014GL062447.

---

## Author Comment (AC2) · 26 Jan 2018

*Responses to Reviewer 2*

*We truly appreciate the sincere point of view of the anonymous Reviewer 2. We found these comments very valuable, insightful and challenging at the same time. We hope we have fulfilled her/his expectations.*

*\*The original comments appeared as one single paragraph in the file that was accessible to us. We separated the paragraph on the main ideas we believe the Reviewer was interested in.*

This manuscript is a case study for reconstruction of streamflow based on tree-ring growth data in Chile. Just to put things in perspective, and I do not mean any negativity here, there is nothing new in this particular manuscript with regard to dendrohydrology. There is almost 50 years of literature in this area; the same regression models, the same way of reconstruction, and the same variable (streamflow). The authors reconstructed summer flows, instead of annual flows, but reconstruction of seasonal variables rather than annual ones happened also several times before but perhaps not in Chile. So, from my point of view, there is not any aspect of novelty in this research. It is important to state this, at least to give the authors a chance to clarify in case I missed something, and I apologize if I did.

**RESPONSE: We truly appreciate your comment because it gives us the opportunity to better stress the importance of our study. This summer reconstruction is important for three main reasons: (1) Chile is one of the countries undergoing strongest precipitation decreases in the last century and (especially the region between 37° and 42°S) it is where there is more agreement among models on further reductions by the end of the 21$^{st}$ century (Koirala et al 2014); understanding how these changes translate into streamflow is needed in order to provide accurate information for developing adaptation and mitigation actions. (2) Although there are many studies about streamflow reconstruction, as the reviewer correctly points out, our work is different and novel relative to those because we explicitly aim to provide an evidence-based criticism of current and proposed water rights regulations and practices. As we posit in the manuscript, the Chilean model is fairly unique and has been studied at the international level for several decades, but to our knowledge this is the first time a dendrohydrological study goes into using this kind of data to discuss the implications for water resources management in detail; we hope that our study encourages other groups to begin a evidence-based discussion on these matters. (3) From a technical standpoint, reconstructing summer streamflow is challenging because relative variations (in percentages) of the average streamflow can be very large at the interannual scale. This way, as the first study focused on this season we believe our work is major step toward understanding base flow dynamics on a multi-century scale.**

The section that is most interesting to read is section 4, which the authors called "Discussion and conclusion". In this section, the authors argued hard for the utility of tree ring-based reconstruction to identify droughts that are more sever or more frequent than those inferred from the instrumental record. This argument can be found in so many of the dendrohydrology papers published in the last decade, and again nothing here is new. However, I have a fundamental issue with the scientific foundation of the argument, and unfortunately this applies also to several other papers published in this field. The authors reconstruct almost 300 years of streamflow data and start comparing it with

instrumental record of 60 years. Ob- viously, they find droughts in the 300 years with characteristics that are different from those in the 60 years, of course! But for water resources engineers, whatever you find in 300-year record MIGHT be a 300-year drought. It is unfair to compare it with the 60-year record. Engineers would fit a statistical distribution to the 60-year record, estimate 100 or 1000-year droughts, then fit a distribution to the longer reconstructed record, and again estimate whatever drought quantiles you want, then compare. Oth- erwise, engineers and water resources planners never use just the deterministic short instrumental record of flows. If you can prove, based on the analysis I suggested, that the reconstructed flows lead to significantly different frequency or severity of droughts, then you made the case about the utility of reconstruction.

**RESPONSE: We agree that there are several papers on reconstructions in different parts of the world. But we insist that there is very few research explicitly proposing approaches for using this information in water management such as our manuscript. In South America, there are only 7 papers published on streamflow reconstruction, a small number considering the large and complex river network across the continent. About the other criticism regarding the unfair comparison between observations and reconstruction, we fully understand the concerns of the reviewer. We want to be very clear here, we are not dismissing the importance of available records and if our writing in some ways suggests that, we assure you it was unintentional. Yes, we agree it is unfair, but it is relevant to keep in mind that the records for the last 60 years have shown important streamflow reduction (e.g. Garreaud et al 2017). We are convinced that our study provides long-term context for these recent fluctuations and our aim is to provide evidence for further discussion in both, the hydroclimatic and water management communities. Thus, our results suggest that the post-1980 period is a fairly unique dry one (for summer) in the context of the last 300 years. In the context of the instrumental record, the post-1980 period represents ~50% of the composite instrumental time-series, while it only represents ~10% of the reconstructed period; we believe this highlights the uniqueness of the post-1980 period.**

Other specific points: Page 2, Lines 8-10: If you have the future projections based on the CMIP5 results, why don't you investigate if projected future droughts are more sever/frequent than the past ones?

**RESPONSE: We appreciate the suggestion but our objective in the paper is to provide context for the low flows that have already occurred. The section you mention is part of our literature review. We haven't utilized CMIP5 projections in this study but it is important to point out that Garreaud (2015) presents a figure where drought recurrence is calculated using CMIP5 output. In that document, it is clear that in the RCP8.5 droughts become more recurrent and that even by the end of the century the definition of drought becomes irrelevant. We will include this explanation in our discussion section.**

Just an idea; Page 3, Line 27: How did you get the "natural regime"? Did you account for irrigation abstraction?

**RESPONSE: Thanks for the suggestion, in the new manuscript we supplement our explanation of natural regime with a text similar to this: In this reconstruction, we selected stations from rivers where the water has not being diverted for irrigation and hydroelectricity. The three selected stations fulfill these criteria. As matter of**

**fact, there are two protected areas here: Reserva Nacional Malalcahuello and Parque Nacional Conguillío.**

Page 3, Line 29: This sentence is not clear. How did you use the double mass curves to determine the calibration window of time?
**RESPONSE: Thanks for catching this up. We believe we need to improve our writing and word choice here in order to explain this idea more clearly. We utilized double mass curves to determine periods that more closely follow precipitation and thus provide us support to detect unreliable records. We eliminated records that didn't fulfill this criterion during the first few years of the time-series.**

Page 4: I feel that I miss proper information about the hydroclimatology of the region. How wet or dry is it? How much is the rain and its variation? Just provide some background in- formation;
**RESPONSE: We appreciate your suggestion. We have included a climograph (Figure R2.A) using data from Temuco for the period 1980-2010. In this figure you can see that January and February get the lowest amounts of rainfall.**

[Figure]

**Figure R2.A: Climograph representing the conditions for Temuco for the period 1980-2010.**

Page 4, Line 17: You are referring to Table 3 before Table 2, please reorder the Tables;
**RESPONSE: We already answered this comment to the Reviewer 1: "We do not agree with this suggestion. Table 2 follows Table 1 in the sense the Table 1 presents the instrumental record and Table 2 provides the information for the analysis of that instrumental record. Then Table 3 appears because it is about the tree-ring chronologies. If we do the change suggested, we feel the manuscript loses readability and that our line of argument weakens".**

Page 4, Lines 24-26: peak over threshold is usually used for floods, but here you are doing drought analysis. Do you mean flow below threshold or something like this? You also need a reference for portion;

**RESPONSE: Thanks for catching this up. We will improve the description in that section. In effect, we use that method (fully described in Mudelsee 2010) to identify percentiles below the 20% during the whole reconstruction. Our objective was to determine how frequent this percentile has been in the reconstruction. This kind of analysis/representation has been utilized in papers analyzing the BioBío river (Muñoz et al 2016), PDSI reconstructions (Christie et al 2011), and in instrumental records (González-Reyes et al 2017).**

Page 5, Line 25: What are these percentage numbers (54.06% and 74.12%)? Do you mean m3/s?

**RESPONSE: These are percentages with respect to the average flow. We have modified the text here in order to make it clear the meaning of these numbers.**

Page 6, Line 2: "that" should be "than", and "here" is not clear. Do you mean your manuscript? If so, why is the reference, it is confusing;

**RESPONSE: Thank you for catching this up. We have modified that section (Lines 1-3) for the following: "Two of three Araucaria araucana tree-ring chronologies extended 800 years or more (PAG and LYV; Fig. 3 and Table 3), which corroborates findings from a previous study (Mundo et al 2012) on the potential of this species for providing long paleoclimatic reconstructions."**

Page 6, Lines 11-17: These results are not really good (especially, RE of 0.36), I know they are typical in many dendrohydrology studies, but they should at least make the authors a bit more humble and lighten the assertion tone that is coming later in Section 4;

**RESPONSE: We agree with this comment. We recognize this value does not look too good; we have modified the text in order to explicitly specify that this results is good in the context of dendrohydrology. The Reduction of Error (RE) accounts for the relationship between the actual value and its estimate. This RE is however good for dendrohydrology studies. A classic paper in this field is Woodhouse (2001) about a reconstruction of streamflow in the Colorado Front Range where the RE was 0.277. In another important paper, Sauchyn et al (2015) report a RE of 0.73 for a reconstruction of the Atabasca River.**

Page 6, Lines 23-24: What does this sentence mean?

**RESPONSE: Thanks for this question. We have modified this sentence in order to show more clearly at what temporal scale the dry years calculated from the instrumental record fit into the reconstructed streamflow: "In order to assess the uniqueness of this recent period of extreme summer streamflow, we (a) divided the tree-ring reconstruction into continuous periods of one, five, 10, and 20 years; and (b) we ranked those periods according with their departure from the mean. According to this classification, the dry period 1996-2000, one of the driest in the instrumental record, ranks fourth in the reconstruction, closely followed by 1987-1991".**

Page 6, Lines 25-27: Is there any meaning for these windows of 5-year, 20-year,..etc? Of course, every time you change the window, you can get different results, but what are we supposed to learn from this?

**RESPONSE: We understand the concern of the reviewer and we clarify this in the new version of the manuscript. Although it is true this time windows are arbitrarily specified, our interest here is to provide context for the occurrence of extreme flows along the whole study period in way that is easier to understand for water managers. Yes, different windows will show different results, but the stress that our intention here is to provide context for the driest summers in the record. This method helps us in perform a more robust comparison between extreme years showing up in the instrumental record and those from the long-term reconstruction. We consider the use of these windows as reliable tool because have allowed us to determine the uniqueness of the post-1980 period.**

Page 6, Line 27: You cannot really use reconstructed flows to comment on extremely high streamflow. Look at your Figure 4 (top left) and you will agree with me;
**RESPONSE: We understand this concern and that is the reason why we briefly describe high flow in that section and instead we focus on low flows. In fact, in the discussion and conclusion section we already had developed a possible explanation for this behavior based upon our results and literature review (Page 9, Lines 7-29). With the new division of sections as requested by the Reviewer 2, we have modified this section in order to better express our ideas.**

Page 6, Lines 29-33: I cannot understand this portion;
**RESPONSE: We concur with the Reviewer this section does not read well. We have modified those lines with the following text "Since 1980, years in the lowest 20$^{th}$ percentile of the reconstruction have become more frequent. We calculated the return period of these low flow years in different periods of the reconstruction. We found (a) that during 1709-1750 and 1940-1960, events with streamflow below the 20th percentile had a 20-year return period; (b) a 5-year return period in 1750-1880; c) a predominantly 2 to 3-year period for 1880-1930; and d) a trend toward a 2-year return period since 1960 (Fig. 5)."**

Page 7, Lines 1-5: What does this argument imply? Rain and streamflow are different! So, how did you conclude that it is a pluvial system? I think you need to elaborate;
**RESPONSE: We appreciate this comment, which allow us to further clarify the correlations we performed in our study. Yes, rain and streamflow are not the same, but it is important to remember that the basin we are analyzing is in a temperate climate. In this region, streamflow data clearly shows that river regime goes from nivo-pluvial regime (high elevation) to pluvio-nival and purely pluvial in lowland areas. Thus, it is expected that streamflow in lower sites are more correlated with rainfall. Thus, the correlation with the rainfall reconstruction validates the record (similarly as for the double mass curves), but it additionally demonstrates the pluvial character of the streamflow at this location and corroborates that our reconstruction is skilled in representing the hydroclimate of the region.**

Pages 7-11: Almost half of the paper came under one section called Discussion and Conclusions. This is a style and format issue that does not look good. You need to include more analysis with the Results section, then not very long Discussion section, then a separate Conclusions section, this will be better.
**RESPONSE: Thanks for this suggestion. We have split discussion and conclusion sections, shortened the former and added some new analysis in the results**

**according to previous suggestions by both reviewers (e.g. IPO analysis and climographs).**

Page 7, Line 31: Usually trend analysis is misleading. Have you looked at the trend of the entire reconstructed record?
**RESPONSE: Thanks for this observation. Earlier in our research we calculated the slope for the entire record and it wasn't statistically significant. We did a new analysis considering the period 1709-2015 (according to other analyses suggested by both Reviewers) and found a non-significant slope of -0.0003. For reference, we will include this value in the first part of the results that describe findings from the tree-ring reconstruction.**

Page 8, Line 5: I got confused, was that SAM work done in this study or taken from other studies?
**RESPONSE: We apologize if this does not read clearly in the manuscript. We will modify it in order to state more clearly that the SAM indexes have been drawn from public sources (e.g. NCAR-NCEP) and databases associated with peer-reviewed publications (e.g. Villalba et al 2012).**

Page 8, Lines 10-12: On what basis was this statement made? Looking at Figure 6 and the correlation numbers does not give me the same impression that the authors have;
**RESPONSE: We understand the concern of the reviewer. We think this is an issue of word choice in our text; we wanted to state that the significant correlation with SAM indicates that our reconstruction captures characteristics of the regional hydroclimate. The use of the article "the" gives the impression that we are claiming that the reconstruction captures "all" features of the regional hydroclimate. We changed that statement for the following: "The tree-ring reconstruction showed a statistically significant correlation with the SAM reconstruction presented in Villalba et al. (2012), especially for the long-term trend; we consider this result confirms that our reconstruction captures characteristics of the regional hydroclimate." We have also modified the caption on Figure 6 to indicate that the asterisks represent statistical significance.**

Page 11, Lines 21-24: I cannot find proof for this in the manuscript, perhaps the authors need to rewrite this;
**RESPONSE: We agree with your assessment. We have deleted this sentence from our manuscript.**

Figure 2 (but also a general comment): On what basis was the selection of Jan-Feb only? Why not March and April too? They also seem part of the low flow season, especially that it may not be a good idea to call two months a drought; The authors also need to note that averaging the streamflow of three stations may reduce the variability in individual gauges, and make the reconstruction easier (nevertheless, the reconstruction accuracy is not high anyways). So, you need to justify this;
**RESPONSE: We selected January and February as representing summer because they the only two months that fall completely in this season. In the Southern Hemisphere the summer runs from December 21$^{st}$ to March 21$^{st}$. I addition, these two months correspond to the lowest rainfall of the year, as shown in the Figure R2.A**
**Concerning the averaging if the individual gauges and the implications for the reconstruction, we agree with the reviewer that averaging the stations may reduce**

variability of individual gauges, but the objective of tree-ring reconstructions is to generate regional (for instance basin-scale) hydroclimatic time-series rather than simulate individual records (gauges). An individual gauge station can be affected by local features, but a composite time-series has more chances to average-out local particularities and thus provide a common regional signal that can be compared with the regional signal of a (or several) tree-ring chronology (ies). Despite this, we consider important to recall Figure R1.A (response to Reviewer 1) where we show that our reconstructed streamflow correlates significantly with observed streamflow of each individual station, and also each individual station correlates significantly with the composite time-series produced in our study.  In addition, in the response to Reviewer 1 we further explain, citing Figure 1, that the reconstruction compares well with the instrumental records utilized for generating the composite.

Table 1: Please report the standard deviation also or better, the coefficient of variation to see the variability of each series;
**RESPONSE: Thank you for this suggestion. We will add the coefficient of variation to that table.**

Table 3: What are those LAN and VILL in the Table title?
**RESPONSE: Thanks for this comment. We added the description of these chronologies in the caption of the new Table 3.**

Table 4: The autocorrelation of the tree ring chronologies are quite high, and is usually transferred to the reconstructed annual flows. I find this unrealistically high for annual flows, can you comment on this and its impact on the reliability of the reconstructed flows? Can you compare it with the autocorrelation of the instrumental flows?
**RESPONSE: We agree with the reviewer and we believe it is important to include information on autocorrelation in the manuscript. Now in the manuscript we include the following text: "The tree-ring reconstruction presents high autocorrelation, consistent with the fact that tree-growth has a temporal memory associated with the water reserve and the soil moisture that remains and is captured by the tree. Some of the statistical procedures applied to the tree-ring chronologies are meant to minimize these effects, but it is virtually impossible to eliminate all. In our case, the autocorrelation is 0.56 to 0.49 for each individual chronology, while it is 0.248 for the reconstruction (1709-2005). This autocorrelation is still high considering that the instrumental record is essentially free from autocorrelation (-0.093)".**

**New References**

**Christie, D.A., Boninsegna, J.A., Cleaveland, M.K., Lara, A., LeQuesne, C., Morales, M.S., Mudelsee, M., Stahle, D.W. & Villalba, R.(2011) Aridity changes in the Temperate-Mediterranean transition of the Andes since AD 1346 reconstructed from tree-rings. Climate Dynamics 36, 1505–1521.**

**Garreaud, R. D., Alvarez-Garreton, C., Barichivich, J., Boisier, J. P., Christie, D., Galleguillos, M., LeQuesne, C., McPhee, J., and Zambrano-Bigiarini, M.: The 2010–2015 megadrought in central Chile: impacts on regional hydroclimate and**

vegetation, Hydrol. Earth Syst. Sci., 21, 6307-6327, https://doi.org/10.5194/hess-21-6307-2017, 2017.

González-Reyes, Á., J. McPhee, D.A. Christie, C. Le Quesne, P. Szejner, M.H. Masiokas, R. Villalba, A.A. Muñoz, and S. Crespo, 2017: Spatiotemporal Variations in Hydroclimate across the Mediterranean Andes (30°–37°S) since the Early Twentieth Century. J. Hydrometeor., 18, 1929–1942, https://doi.org/10.1175/JHM-D-16-0004.1

Koirala, S., Hirabayashi, Y., Mahendran, R., & Kanae, S. (2014). Global assessment of agreement among streamflow projections using CMIP5 model outputs. Environmental Research Letters, 9(6): 064017. doi:10.1088/1748-9326/9/6/064017.

Mudelsee M (2010) Climate Time Series Analysis: Classical Statistical and Bootstrap Methods. Springer, Dordrecht.